

# Three new species of the family Idyanthidae (Copepoda, Harpacticoida) from sublittoral zones around the Korean Peninsula

Jong Guk Kim[1], Kyuhee Cho[2] and Jimin Lee[2]

[1] Division of Zoology, Honam National Institute of Biological Resources, Mokpo, Republic of Korea
[2] Ocean Climate Response & Ecosystem Research Department, Korea Institute of Ocean Science & Technology, Busan, Republic of Korea

Corresponding author
Jimin Lee, leejm@kiost.ac.kr

## ABSTRACT

There are few taxonomic studies of the harpacticoid family Idyanthidae Lang, 1944 in the Pacific Ocean. As a first report of the family in this region, one species of the monotypic genus *Idyellopsis* Lang, 1948 and two species of the genus *Idyella* Sars, 1905 are described from sublittoral habitats around the Korean Peninsula. *Idyellopsis orientalis* sp. nov. is distinguished from the type and only species of the genus, *Idyellopsis typica* Lang, 1948, in the length:width ratio of the body and genital double-somite, length of caudal seta I, and structure of the female P5 baseoendopod. *Idyella dolichi* sp. nov. and *Idyella exochos* sp. nov. share the armature of the female P5 exopod with four setae with *Idyella exigua* Sars, 1905, *Idyella major* Sars, 1920, and *Idyella pallidula* Sars, 1905. However, the two new species differ from the three European species in the armature of the terminal segment of the P1 endopod and female P5 baseoendopod, and in the shape of the genital double-somite. The male of *Idyella dolichi* sp. nov. also has two distinct features: the distal two segments of the P2 endopod are completely fused, and the third segment of the antennule has a prominent outer projection. The geographical distribution of *Idyellopsis* and *Idyella* is extended by the discovery of three new species in Korean waters. We also discuss a possible phylogenetic relationship among members of family Idyanthidae at genus level.

## INTRODUCTION

The benthic harpacticoid family Idyanthidae Lang, 1944 is a small harpacticoid taxon comprising 25 species in 10 genera, most of which are monotypic except for *Idyella* Sars, 1905 (seven species), *Tachidiella* Sars, 1909 (six species), *Idyanthe* Sars, 1909 (four species), and *Pseudometeorina* George & Wiest, 2015 (two species) (*Wells, 2007*; *George, 2023*). *Seifried (2003)* established the monophyly of this family by excluding it from the family Tisbidae Stebbing, 1910 *sensu stricto* and raised it to full family status, establishing Idyantidimorpha Seifried, 2003 composed of Idyanthidae and Zosimeidae Seifried, 2003. Subsequently, *Moura & Martínez Arbizu (2003)* formally synonymized the deep-sea family

Styracothoracidae Huys, 1993 comprising only *Styracothorax gladiator* Huys, 1993, with Idyanthidae. This taxonomic decision led to the allocation of the deep-sea taxa *Aspinothorax* Moura & Martínez Arbizu, 2003, *Meteorina* George, 2004, and *Pseudometeorina* into the family, but their taxonomic positions remained complicated until *George*'s *(2023)* contribution (*George & Wiest, 2015*). *George (2023)* provided a novel insight into their status and established a new subfamily Aspinothoracinae George, 2023 for these four genera.

Idyanthid copepods occur in sublittoral substrata from shallow waters to the abyssal deep sea (*Huys, 1993*; *Lee & Huys, 1999*; *George & Wiest, 2015*); all members of the subfamily Aspinothoracinae are known from deep-sea habitats. They have been mainly recorded in European seas where the diversity of harpacticoid copepods has been well studied, and from the Antarctic and Arctic Oceans (*Lee & Huys, 1999*; *Moura & Martínez Arbizu, 2003*). There are a few taxonomic studies on idyanthid copepods in the Pacific Ocean. *Lang (1965)* reported *Tachidiella parva* Lang, 1965 from Monterey Bay off the Californian Pacific Coast; *Becker (1974)* described the deep-sea species *Dactylopia peruana* Becker, 1974, from the Peru–Chile Trench at a depth of 920 m; and *Huys (1993)* discovered *Styracothorax gladiator* from a depth of 2,050 m northwest of Manila off the Philippines. However, several unidentified species of the family Idyanthidae have been reported in the northwestern Pacific Ocean, including ecological records of two species of the genus *Idyellopsis* Lang, 1948 from a bathyal site in Sagami Bay off central Japan (*Shimanaga, Kitazato & Shirayama, 2004*), three genera *Idyellopsis*, *Nematovorax* Bröhldick, 2005 and *Tachidiella* from Kuril Trench (*Kitahashi et al., 2013*), and the genus *Nematovorax* from Ryukyu Trench (*Kitahashi et al., 2014*).

During a study of the meiofaunal community in Korean waters, one species of the monotypic genus *Idyellopsis* and two species of the genus *Idyella* were found in sublittoral habitats. In the present study, we describe these three species as the first report of the family Idyanthidae from this region, extending the geographical distributions of both genera.

## MATERIALS AND METHODS

Benthic samples of various substrate types were obtained with an on-board Smith-McIntyre grab sampler (0.1 m$^2$) at depths of 50–110 m, and aided by divers at depths of 20–30 m. Figure 1 shows the sampling stations for three new Idyanthidae species. Each sample was sieved through a hand net with a 50-μm mesh size and fixed in 10% formalin or immediately preserved in 95% ethanol. In the laboratory, harpacticoid copepods were sorted using a Leica M165 C stereomicroscope; the material examined in this study was prepared in lactic acid for light microscopy observation. The three new idyanthid species were drawn using a drawing tube attached to differential interference contrast microscopes (Olympus BX53 or Leica DM2500). Dissected specimens were mounted in lactophenol: glycerin (1:5) on H-S slides (*Shirayama, Kaku & Higgins, 1993*) and sealed with nail varnish; other specimens were preserved in a vial with 80% ethanol. The type series of *Idyellopsis orientalis* sp. nov. and *Idyella dolichi* sp. nov. are in the Marine Biodiversity Institute of Korea, Seocheon, Korea, and the type specimens of *Idyella exochos* sp. nov. are in the Honam National Institute of Biological Resources, Mokpo, Korea.

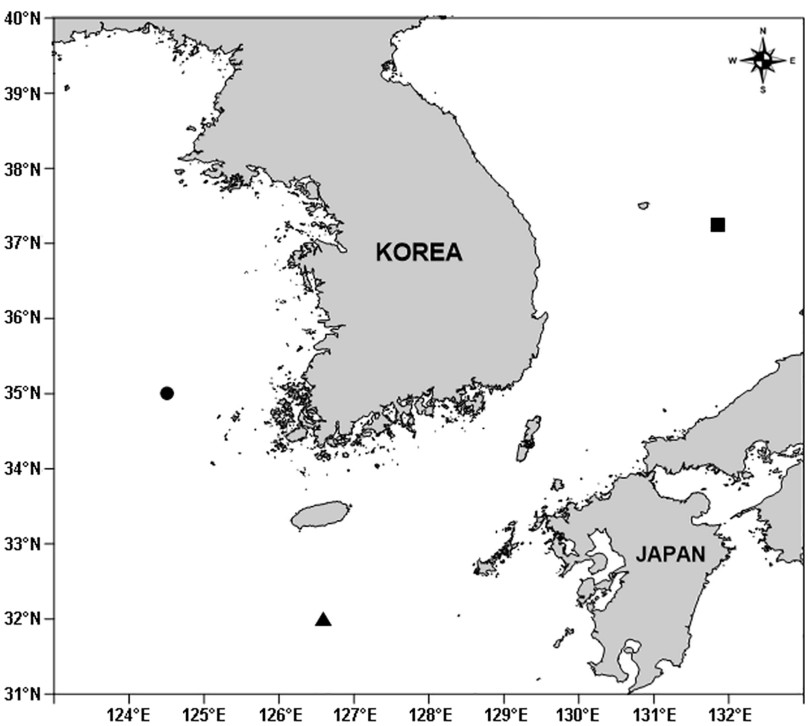

**Figure 1 A map showing sampling stations of three new idyanthid species around the Korean Peninsula.** Filled circle, *Idyellopsis orientalis* sp. nov.; filled triangle, *Idyella dolichi* sp. nov.; filled square, *Idyella exochos* sp. nov.

The terminology for morphological descriptions follows that of *Huys & Boxshall (1991)*, and the setal armature formulae of thoracic legs are adopted from *Lang (1934)*. Abbreviations in the text, figures, and tables are: ae, aesthetasc; apo, apophysis; exp, exopod; enp, endopod; exp(enp)-1(-2, -3) to denote the proximal (middle, distal) segment of a ramus; and P1–P6, first to sixth thoracopods.

The electronic version of this article in Portable Document Format (PDF) will represent a published work according to the International Commission on Zoological Nomenclature (ICZN), and hence the new names contained in the electronic version are effectively published under that Code from the electronic edition alone. This published work and the nomenclatural acts it contains have been registered in ZooBank, the online registration system for the ICZN. The ZooBank LSIDs (Life Science Identifiers) can be resolved and the associated information viewed through any standard web browser by appending the LSID to the prefix http://zoobank.org/. The LSID for this publication is: [urn:lsid:zoobank.org: pub: 311543E0-7EBE-4FE5-BC2A-C56B047172A2]. The online version of this work is archived and available from the following digital repositories: PeerJ, PubMed Central and CLOCKSS.

## RESULTS

**Taxonomy**

Order Harpacticoida Sars, 1903

Family Idyanthidae Lang, 1944

**Table 1 Armature formula of P1–P4 in the genus *Idyellopsis* Lang, 1948.**

|     | Exopod  | Endopod            |
| --- | ------- | ------------------ |
| P1  | 0.1.123 | 1.120              |
| P2  | 1.1.223 | 1.2.221 (1.41apo in ♂) |
| P3  | 1.1.323 | 1.1.321            |
| P4  | 1.1.323 | 1.1.221            |

Genus *Idyellopsis* Lang, 1948

Diagnosis.—Idyanthidae. Habitus pyriform, slightly depressed, with clear podoplean boundary between prosome and urosome. P1-bearing somite completely incorporated into cephalothorax. P5-bearing somite small, trapezoidal in dorsal view. Genital somite and third urosomite completely fused ventrally forming genital double-somite in female but partially separated by dorsolateral surface ridges; without lateral extentions in both sexes. Anal somite small, with semicircular operculum. Caudal rami short, as long as wide, with seven setae.

Rostrum small, fused to cephalic shield basally, bent ventrally. Antennule slender, 9-segmented. Antenna 4-segmented; basis without abexopodal seta; exopod 3-segmented, with armature formula 2.1.2; endopod 2-segmented; proximal segment with 1 seta; distal segment with 2 (*I. typica*) or four lateral setae (*I. orientalis* sp. nov.) and seven distal elements. Mandibular palp biramous; exopod 2-segmented; endopod elongate, 1-segmented with one (*I. orientalis* sp. nov.) or two lateral setae (*I. typica*). Maxillulary basis with one lateral endite; exopod and endopod 1-segmented. Maxillary syncoxa with three endites; praecoxal endite bilobed, each with two (*I. orientalis* sp. nov.) or three (*I. typica*) setae; endopod 1-segmented (*I. typica*) or 3-segmented (*I. orientalis* sp. nov.). Maxilliped subchelate, outwardly flexible at joint between syncoxa and basis; syncoxa with two setae; basis with one seta on straight inner margin, outer margin convex; endopod 2-segmented, proximal segment with one claw and one seta and distal segment with two long setae and one short seta (but absent in *I. typica*). P1 with 3-segmented exopod and 2-segmented endopod; enp-1 elongate and broadened at level of inner seta; enp-2 with three setae. P2–P4 with 3-segmented rami; P2 enp-2 with two inner setae, P3–P4 with one inner seta. P5 exopod and baseoendopod separate; endopodal lobe well-developed, with three setae; exopod elongate, with four setae. P6 represented by small plate bearing two setae. Armature formula of P1–P4 given in Table 1.

Sexual dimorphism observed in the male urosome, antennule, P2 endopod, P5, and P6. Urosome 6-segmented; genital somite and third urosomite separate. Antennule subchirocer, 8- (in *I. typica*) or 9-segmented. P2 endopod 2-segmented; original second and third segments with division visible only from posterior view; enp-3 with two plumose inner setae, one modified small apical seta and one very reduced outer apophysis. P5 exopod with four setae and endopodal lobe with two setae. P6 asymmetrical, represented by large plate bearing one seta and two spines.

Type species.—*Idyellopsis typica* Lang, 1948 [by original designation]

Other species.— *Idyellopsis orientalis* sp. nov.

*Idyellopsis orientalis* sp. nov.
urn:lsid:zoobank.org:act:9998A5DB-A356-43A8-8628-5AFBD9E369EE
Figures 2–5

Type locality.—Yellow Sea, 34°59′56.50″N, 124°30′00.18″E; 89.5 m depth.

Type material.—Holotype: ♀ (MABIK CR00257810) preserved in a vial with 80% ethanol; collected from the type locality, 02 March 2022. Paratypes: 2 ♀♀ (MABIK CR00257807, CR00257809) dissected and mounted on 10 or 3 H-S slides; 1 ♂ (MABIK CR00257808) dissected and mounted on 4 H-S slides; collection data as in holotype; coll. J. G. Kim.

Description of female (based on the holotype and paratypes).—Holotype body length 358 μm (figured paratype: 342 μm), measured from anterior margin of cephalothorax to posterior margin of caudal rami in dorsal view.

Habitus (Fig. 2A) pyriform, depressed, with clear separation between prosome and urosome. Prosome (Fig. 2A) slightly longer than urosome, comprising cephalothorax with completely fused first pedigerous somite and three free pedigerous somites, ornamentation as figured. Cephalothorax slightly wider than long, about 34% body length.

Urosome (Figs. 2A and 2F) comprising P5-bearing somite, genital double-somite, and three free abdominal somites, ornamentation as figured. P5-bearing somite smallest, dorsally with one anterior pore and one pair of posterior sensilla. Genital somite and first abdominal somite fused forming genital double-somite, original genital segmentation marked by distinct dorsolateral surface ridge. Genital field located ventrally in the middle of anterior half of somite, with a small copulatory pore, and P6 (Fig. 2G) represented by small plate bearing two bare setae; copulatory pore located in medial depression (Fig. 2F, figured from other paratype). Anal somite small, ventrally with pairs of 2 large ventral and three small ventrolateral pores; ornamented ventrally with one pair of spinular rows and one pair of posterolateral spinular rows, and dorsally with one pair of sensilla; operculum semicircular, wide, located at anterior 1/3 of anal somite.

Caudal rami (Figs. 2A, 2F, and 2H) slightly longer than wide, ornamented with three rows of minute dorsal spinules, one row of stout ventral spinules, and one ventral large pore; with seven setae: setae I and II inserted in proximal half of ramus; seta I smallest, seta II 2 times longer than seta I, seta III about three times longer than seta I; principal setae IV and V fused basally; seta IV unipinnate and seta V about two times longer than seta IV; seta VI as long as seta II, uni-pinnate; seta VII tri-articulate, inserted in proximal 2/3 of ramus on dorsal surface.

Rostrum (Fig. 2B) completely fused to cephalothorax, ventrally deflected, invisible in dorsal view, with paired sensilla, with pointed tip.

Antennule (Fig. 3A) 9-segmented. First segment with inner spinules. Second segment longest. Fourth segment with one peduncle carrying one aesthetasc fused to one long seta at distal inner corner. Seventh segment shortest, with two bi-articulate and two simple setae. Distal segment with three (2 long setae + 1 rudimentary element indicated by arrowhead in Fig. 3A) and four bi-articulate setae, and one small aesthetasc fused to one

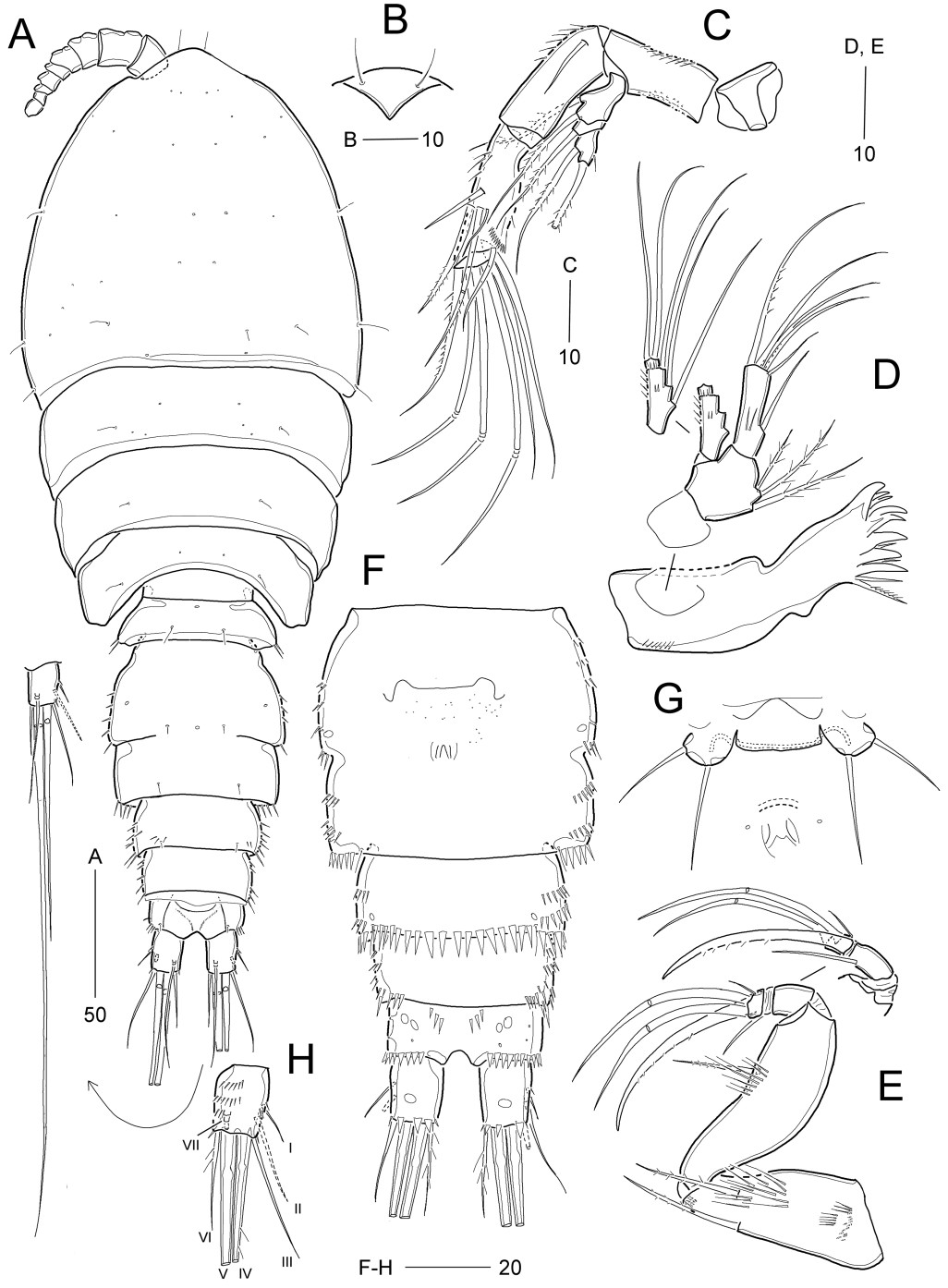

**Figure 2** *Idyellopsis orientalis* **sp. nov., female holotype (MABIK CR00257810, A and B) and paratypes (MABIK CR00257807, C–F, H; MABIK CR00257809, G).** (A) Habitus, dorsal; (B) rostrum, ventral; (C) Antenna; (D) mandible; (E) maxilliped; (F) urosome excluding P5 bearing-somite, ventral; (G) P6; (H) caudal ramus, dorsal. Scale bars are in μm.

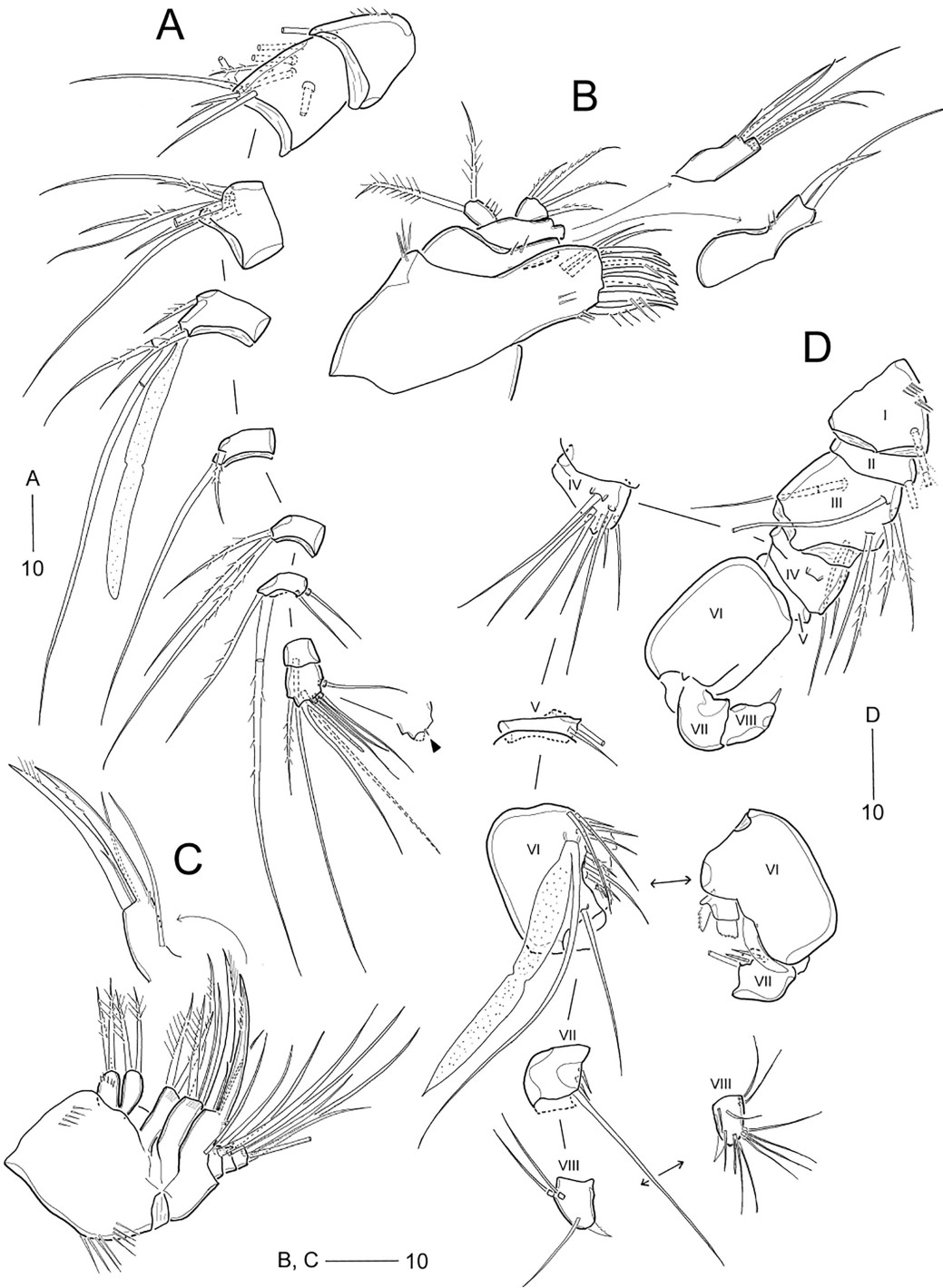

**Figure 3** *Idyellopsis orientalis* **sp. nov., female paratype (MABIK CR00257807, A–C) and male paratype (MABIK CR00257808, D).** (A) Antennule, arrowhead indicating a rudimentary element on the distal segment; (B) maxillule; (C) maxilla; (D) antennule, other side of sixth and eighth segments figured additionally. Scale bars are in μm.

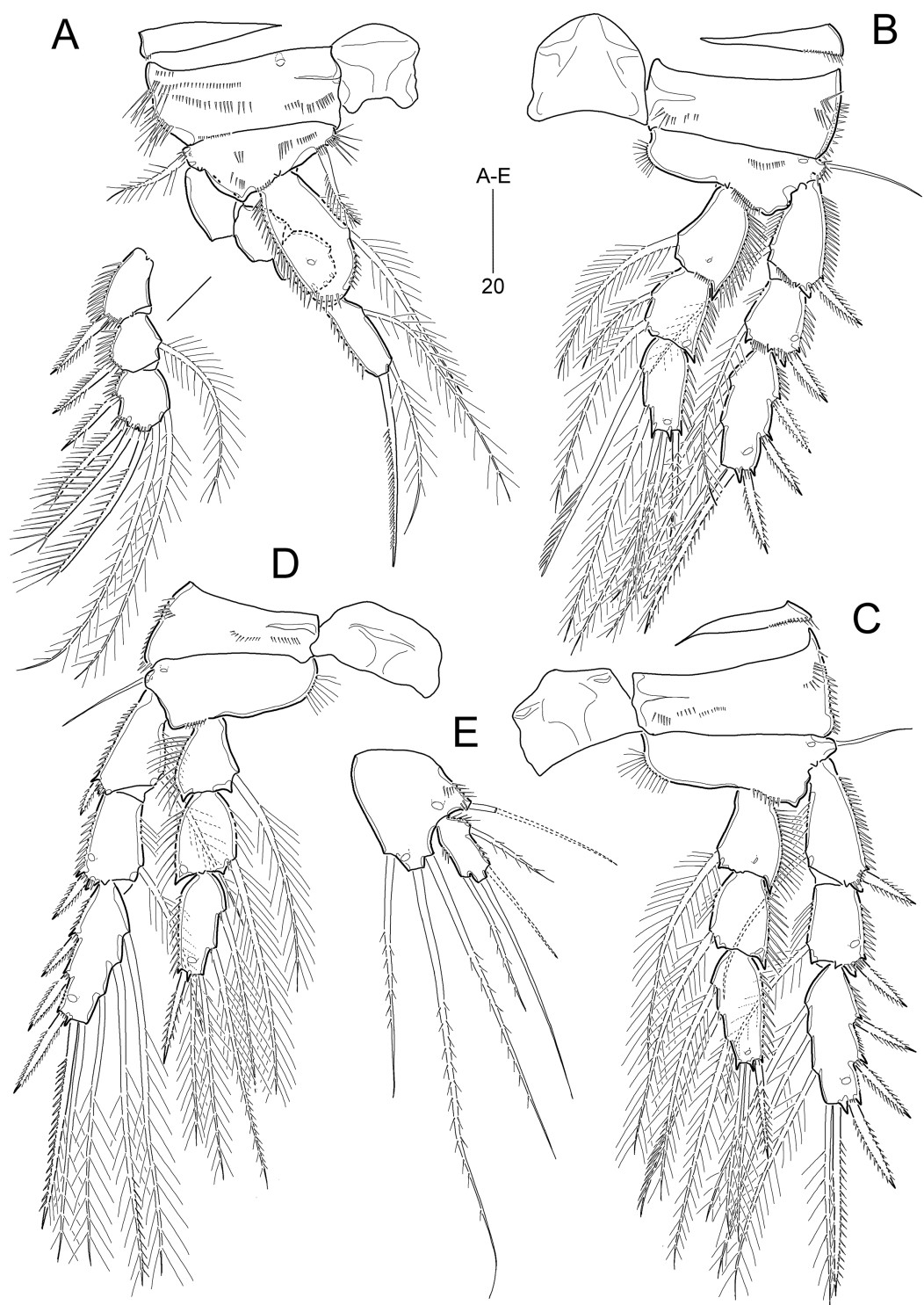

**Figure 4** *Idyellopsis orientalis* **sp. nov., female paratype (MABIK CR00257807).** (A) P1, anterior; (B) P2, anterior; (C) P3, anterior; (D) P4, anterior; (E) P5. Scale bars are in μm.

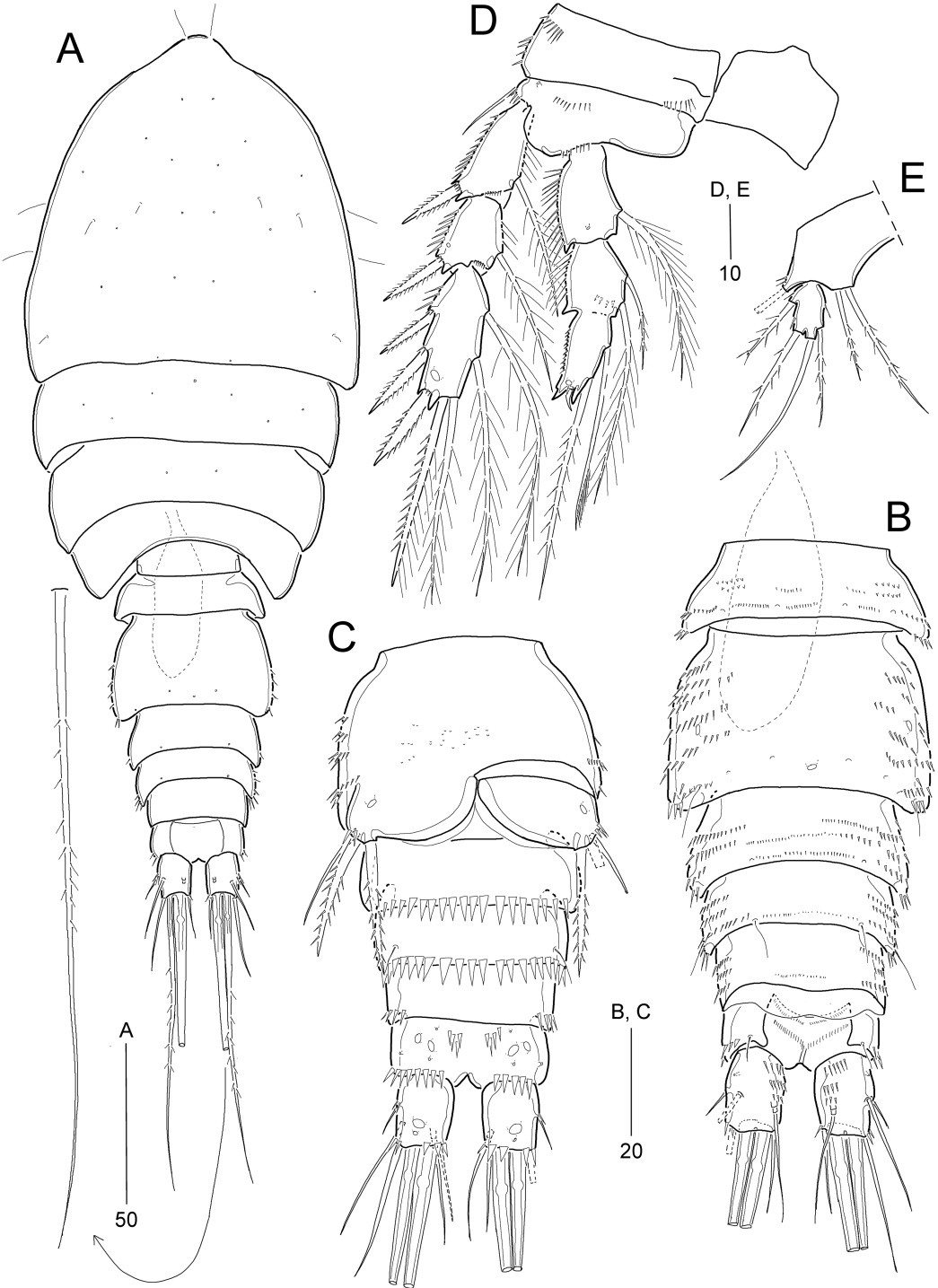

**Figure 5 _Idyellopsis orientalis_ sp. nov., male paratype (MABIK CR00257808).** (A) Habitus, dorsal; (B) urosome, dorsal; (C) urosome excluding P5 bearing-somite, ventral; (D) P2; (E) P5. Scale bars are in µm.

long seta. Setal formula: 1-[1], 2-[9], 3-[7], 4-[3 + (1 + ae)], 5-[2], 6-[3], 7-[4], 8-[1], 9-[7 + (1 + ae)].

Antenna (Fig. 2C) 4-segmented, comprising coxa, basis, and two endopodal segments. Coxa small, unornamented. Basis elongate, ornamented with inner spinules and outer minute denticles. Exopod arising distally on basis, 3-segmented; proximal segment longest, with two lateral setae; middle segment smallest, with one lateral seta; distal segment with one distal and one lateral seta, ornamented with one lateral spinule. Endopod 2-segmented; proximal segment slightly longer than basis, with one abexopodal seta, and proximally ornamented with inner spinules; distal segment as long as preceding segment, ornamented with inner spinules and outer subdistal hyaline frill, and with four lateral setae; distal armature composed of seven apical setae (one pinnate, three bare, and three geniculate), of which two adjacent outer apical long setae fused at base.

Mandible (Fig. 2D). Coxa long, gnathobase bearing one small dorsal bulge; distal margin with one bi-, one tri-, and three uni-cuspidate teeth, three thin spines and dorsodistal corner with one pinnate seta. Palp well-developed, biramous, consisting of basis, exopod and endopod. Basis broad, with three lateral setae. Exopod 2-segmented, slightly shorter than endopod; proximal segment elongate, with three inner setae, ornamented with few spinules; distal segment small, with two apical setae. Endopod 1-segmented, with one lateral and five apical setae, ornamented with few spinules on surface.

Maxillule (Fig. 3B). Praecoxa with one group of outer spinules; arthrite well-developed, with two juxtaposed anterior setae and eight distal elements (of which three posterior ones bearing one or two spinules, most dorsal one spinulose), and ornamented with few posterior and dorsal spinules. Coxa with cylindrical endite bearing two setae and ornamented with one dorsal and two ventral spinules. Basis with two endites; ventral endite with one small and two long setae; distal endite with three long setae. Exopod 1-segmented, small, with two pinnate distal setae, and ornamented with inner spinules. Endopod 1-segmented, with four pinnate distal setae.

Maxilla (Fig. 3C). Syncoxa large, ornamented with one diagonal row of long spinules on outer margin and one row of minute spinules on anterior surface; with three endites: proximal endite bilobed, each lobe with two pinnate setae distally, and proximal lobe with one row of anterior spinules; middle endite elongate, with two pinnate setae distally and one row of posterior spinules subdistally; distal endite also elongate, with one bare and two pinnate setae distally, and one row of posterior spinules subdistally. Allobasis drawn out into a curved claw accompanied by one tube pore, with one pinnate spine and two bare (one small, one long) setae distally, and one long bare seta proximally. Endopod small, 3-segmented; first and second segments with two subdistal setae each; distal segment with four distal setae.

Maxilliped (Fig. 2E) 4-segmented, composed of syncoxa, basis, and 2-segmented endopod. Syncoxa elongate, with rows of various-sized spinules on anterior and posterior surfaces; with two pinnate setae at 2/3 of segment. Basis longer than syncoxa, inner margin slightly concave and outer margin convex; with one row of long spinules and one pinnate seta on middle of inner margin. Proximal endopodal segment slightly longer than distal

endopodal segment, distally with one claw and one seta; distal endopodal segment small, with one minute lateral seta and two long distal setae with fracture planes.

P1 (Fig. 4A) with bare intercoxal sclerite. Praecoxa transversely elongated, triangular, ornamented with outer spinules on distal margin. Coxa large, with one anterior proximal pore and several rows of various-sized spinules. Basis smaller than coxa, with one anterior pore at base of outer seta, and several rows of various-sized spinules; with one plumose outer and one pinnate inner seta. Exopod 3-segmented, inwardly bent; each segment ornamented with outer spinules; exp-1 and exp-2 each with one bi-pinnate outer spine; exp-2 smallest, with one plumose inner seta; exp-3 with three outer spines (one pinnate and two uni-spinulose), two apical elements (one uni-spinulose spine and one plumose seta), and one plumose inner seta. Endopod longer than exopod, 2-segmented; proximal segment slightly exceeding distal end of exopod, with one anterior pore and one long plumose inner seta; outer margin ornamented with spinules, and inner margin convex ornamented with setules proximally and with spinules subdistally; distal segment as long as 2/3 length of preceding segment, with two apical elements (one uni-pinnate spine and one plumose seta) and one plumose inner seta.

P2–P4 (Figs. 4B–4D). Intercoxal sclerite well-developed and unornamented. Praecoxa transversely elongated (of P4 not figured), ornamented with distal spinules. Coxa large, with several rows of anterior and outer spinules. Basis with one bare outer seta; inner margin rounded, bearing setular tuft, anterior surface with one pore near base of outer peduncle, and distal margin with one row of spinules near base of endopod. Exopod 3-segmented, longer than endopod, exp-1 and exp-2 each with one pinnate outer spine and one inner seta, ornamented with outer spinules and inner setules as figured (Figs. 4B–4D); exp-2 and exp-3 each with one anterior pore near outer distal corner; exp-3 longest, with three pinnate outer spines, two apical elements and two (P2) or three (P3 and P4) plumose inner setae. Endopod 3-segmented, each segment with spinules along outer margin and one anterior pore (except for P4 enp-1); enp-1 with anterior surface pore in P2 and P3 or without pore in P4, with one plumose inner seta; enp-2 ornamented with inner setules, and with one (P3 and P4) or two (P2) plumose inner setae; enp-3 longest, with one pinnate outer spine, two plumose apical setae, and two (P2 and P4) or three (P3) plumose inner setae. Armature formula of P1–P4 given in Table 2.

P5 (Fig. 4E) with separate exopod and baseoendopod. Exopod about 2.2 times as long as broad, ornamented with spinules on outer and inner margins; with two pinnate (?) outer setae, one bare apical seta (longest), and one pinnate inner seta. Baseoendopod with two pores on anterior surface and three rows of spinules near outer margin, with one outer basal seta; endopodal lobe reaching middle of exopod, with three pinnate setae (of which middle one longest).

Description of male.—Body length 268.6 μm in dorsal view. Habitus (Fig. 5A) as in female except for urosomal segmentation with separated second and third urosomites. Sexual dimorphism also observed in antennule, P2, P5, and P6.

Urosome (Figs. 5B and 5C) 6-segmented, with ornamentation as figured. Urosomites, except for anal somite, ornamented with several rows of spinules on dorsal and

**Table 2 Armature formula of P1–P4 of *Idyellopsis orientalis* sp. nov.**

|  | Exopod | Endopod |
|---|---|---|
| P1 | 0.1.123 | 1.120 |
| P2 | 1.1.223 | 1.2.221 (1.41apo in ♂) |
| P3 | 1.1.323 | 1.1.321 |
| P4 | 1.1.323 | 1.1.221 |

dorsolateral surface. Second urosomite with three pores on dorsal surface and minute spinules midway of ventral surface. Third and fourth urosomites each with one row of stout distal spinules ventrally. Anal somite and caudal rami (Figs. 5B and 5C) as in female.

Antennule (Fig. 3D) 8-segmented, subchirocer, with geniculation between sixth and seventh segments. First segment with two rows of inner spinules. Fifth segment smallest. Sixth segment swollen, with three modified elements (one curved spine-like, two short serrate spines). Distal segment conical, slightly curved inwards. Armature formula as follows: 1-[1], 2-[1], 3-[8], 4-[7], 5-[2], 6-[10 + 3 modified elements + (1 + ae)], 7-[3], 8-[10 + 1 modified element (small distal spine) + (1 + ae)].

P2 (Fig. 5D). Exopod as in female. Endopod 2-segmented, original second and third segments fused, but with scar indicating original division posteriorly; enp-1 as in female; original enp-2 as in female, ornamented with anterior spinules; original enp-3 with 2 plumose inner setae, one modified small apical seta, and one very reduced outer apophysis (Table 2).

P5 (Fig. 5E). Baseoendopods of right and left legs fused medially, with one outer basal seta; endopodal lobe with two pinnate distal setae, of which inner one longer. Exopod 1-segmented, smaller than in female, 1.5 times as long as broad, with two pinnate outer setae, 1 bare apical seta (longest), and one pinnate inner seta.

P6 (Fig. 5C). Sixth pair of legs asymmetrical (fused to ventral wall of supporting somite on one side, represented on both sides by a large plate); outer distal corner with one bare seta and two pinnate spines. Distal margin of both plates forming a hyaline membrane.

Etymology.—The name of the new species, *orientalis*, refers to the first record of the genus *Idyellopsis* in East Asia.

Remarks.—*Lang (1944)* established the so far monotypic genus *Idyellopsis* for his newly proposed new species, *Idyellopsis typica* Lang, 1944, but he omitted the description of the species, rendering the latter unavailable. Subsequently, *Lang (1948)* made the name available by adding a brief diagnosis for the species based on female specimens discovered in muddy sediments from Gullmarn Fjord, Sweden, at depths of 50–110 m. In his description of the species, he showed the maxilliped, P1, and female P5 (*Lang, 1948*: 414, Abb. 181(2)).

*Gee & Fleeger (1986)* re-examined Lang's material (one female specimen) deposited in the Stockholm Museum, Sweden, and made three corrections to the original description: the 9-segmented condition of the antennule; the antennary basis lacking any setae; and the 3-segmented condition of the antennary exopod. Based on the re-examination, they

confirmed the identity of the specimens collected from fine sandy sediments of Borge Bay, Signy Island, Antarctica, at a depth of 15 m, providing reliable descriptions of both sexes of *I. typica*. They were aware of the differences in the total body length between the two populations [600–700 μm (♀) in the Swedish population *vs.* 330–370 μm (♀), 280–330 μm (♂) in the Antarctic population] and their different geographical distributions. Given the morphological information reported by *Gee & Fleeger (1986)*, the amended generic boundary of females of *Idyellopsis* is based on the following characteristics: the rostrum is small, fused to cephalic shield; the genital somite and third urosomite (genital double-somite) are completely fused dorsally (without a dorsal suture), and their epimeral plates are weakly developed (not extended as wing-shaped processes); the caudal rami are short, as long as wide; the first antennules are short, 9-segmented, without plumose setae; the antenna has an abexopodal seta on the basis and a 3-segmented exopod; the exopod of P1 has one inner seta on the middle segment and six setae on the terminal segment; the endopod of P1 is 2-segmented, bearing three (one inner, two terminal) setae on the distal segment; the second endopodal segments of P2 and P3 have two and one inner seta, respectively; the exopod of P5 is small, armed with four setae; and the baseoendopod is distally produced, with three setae.

We found a member of the family Idyanthidae in sublittoral muddy sediments in the Yellow Sea of Korea. Morphological examination of the Korean material confirms that this species can be assigned to *Idyellopsis* based on the generic diagnosis mentioned above, with numerous differences that make it recognizable as the second species of *Idyellopsis*. The main differences between the two species in females include the length:width ratio of the body (2.0 in *I. typica vs.* 2.7 in *I. orientalis* sp. nov.) and genital double-somite (0.6 in *I. typica vs.* 0.9 in *I. orientalis* sp. nov.), the length of caudal seta I (as long as seta II in *I. typica vs.* half-length of seta II in *I. orientalis* sp. nov.), the extrusion of the P5 baseoendopod (reaching distal third of exopod in *I. typica vs.* extending only to the proximal third of exopod in *I. orientalis* sp. nov.), and the relative length of the two setae representing P6 (outer seta twice as long as inner one in *I. typica vs.* inner seta slightly longer than outer one in *I. orientalis* sp. nov.).

Additional differences were observed in the setation of cephalosomal appendages in the new species, *i.e.*, the distal endopodal segment of the antenna with four lateral setae (*vs.* two setae in *I. typica*), the basis of mandible with three setae (*vs.* four setae in *I. typica*), the endopod of mandible with one lateral seta (*vs.* two setae in *I. typica*), the exopod of maxillule with two setae (*vs.* three setae in *I. typica*), the endopod of maxillule with four setae (*vs.* six setae in *I. typica*), each lobe of proximal endite of maxilla with two setae (*vs.* each with three setae in *I. typica*), and the distal endopodal segment of maxilliped with two distal setae and an additional lateral seta (*vs.* only two distal setae in *I. typica*). In addition, the maxilla of *I. orientalis* sp. nov. has a 3-segmented endopod whereas the re-description of *I. typica* indicates it is 1-segmented. However, pending a thorough re-examination of the Swedish and Antarctic materials of *I. typica*, these microcharacters may not be sufficient to distinguish the new species from *I. typica*, as older descriptions of harpacticoids often contain several observational errors in the details of mouthparts (*cf.*, *Huys & Mu, 2021*; *Mathiske et al., 2021*; *etc.*).

Although idyanthid copepods show sexual dimorphism in the distal endopodal segment of P2, it is variable among idyanthid genera (*Seifried, 2003*; *Bröhldick, 2005*); the sexual dimorphism of this ramus is often lost in the genus *Pseudometeorina* (*George & Wiest, 2015*; *George, 2023*). According to *Seifried*'s *(2003)* phylogenetic study, this male segment primarily lacks inner elements (lost two inner setae of female) and has a modified spine fused basally to the segment, a hyaline seta, and an inner terminal seta. In the male of *I. typica*, however, this ramus is characterized by retaining two inner setae on the distal segment, lacking the hyaline seta, and somewhat fused second and third segments (see *Gee & Fleeger, 1986*: Fig. 11C). The fact that the new species shares these two sexual dimorphic features with *I. typica* can be considered synapomorphic evidence supporting its generic assignment. Nevertheless, there are significant differences between *I. typica* and the new species in the male: in the new species the antennules are 8-segmented with only two segments distal to the geniculation; the apophysis (derived from the modified outer spine) and inner distal seta on the distal endopodal segment of P2 are remarkably reduced; the second and third endopodal segments of P2 are completed fused; and both baseoendopods of P5 are fused medially. We suggest that the nature of the sexual dimorphism in P2 might be useful for species discrimination in the genus.

Genus *Idyella* Sars, 1905
*Idyella dolichi* sp. nov.
urn:lsid:zoobank.org:act: 5487A79B-3A11-4622-82AD-0A731A91CEBF
Figures 6–9

Type locality.—South Sea, Korea, 31°59′52.41″N, 126°35′34.44″E, 102.8 m depth.

Type material.—Holotype: ♀ (MABIK CR00257806) dissected and mounted on 12 H-S slides; collected from the type locality, 07 August 2016. Paratypes: 1 ♀ (MABIK CR00257805) preserved in a vial with 80% ethanol; 1 ♂ (MABIK CR00257804) dissected and mounted on 3 H-S slides; coll. S. L. Kim.

Description of female (based on holotype and paratype).—Holotype body length 330 μm, measured from anterior margin of cephalothorax to posterior margin of caudal rami in dorsal view.

Habitus (Fig. 6A) broad, pyriform depressed, with distinction between prosome and urosome. Prosome about 1.6 times longer than urosome, comprising cephalothorax and three free pedigerous somites; first pedigerous somite completely fused to cephalosome. Cephalothorax about 40% of body length.

Urosome (Figs. 6A and 6C) comprising P5-bearing somite, genital double-somite, and three free abdominal somites, ornamentation as figured. P5-bearing somite smallest, with spinules on both sides of distal margin and dorsal surface. Genital somite and third urosomite completely fused forming genital double-somite; original genital somite with one pair of backwardly curved posterolateral corniform processes; lateral and distal margins ornamented with spinules, and surface with one median pore subdistally and several pairs of setules. Genital field (Fig. 6C) located anteriorly on genital double-somite,

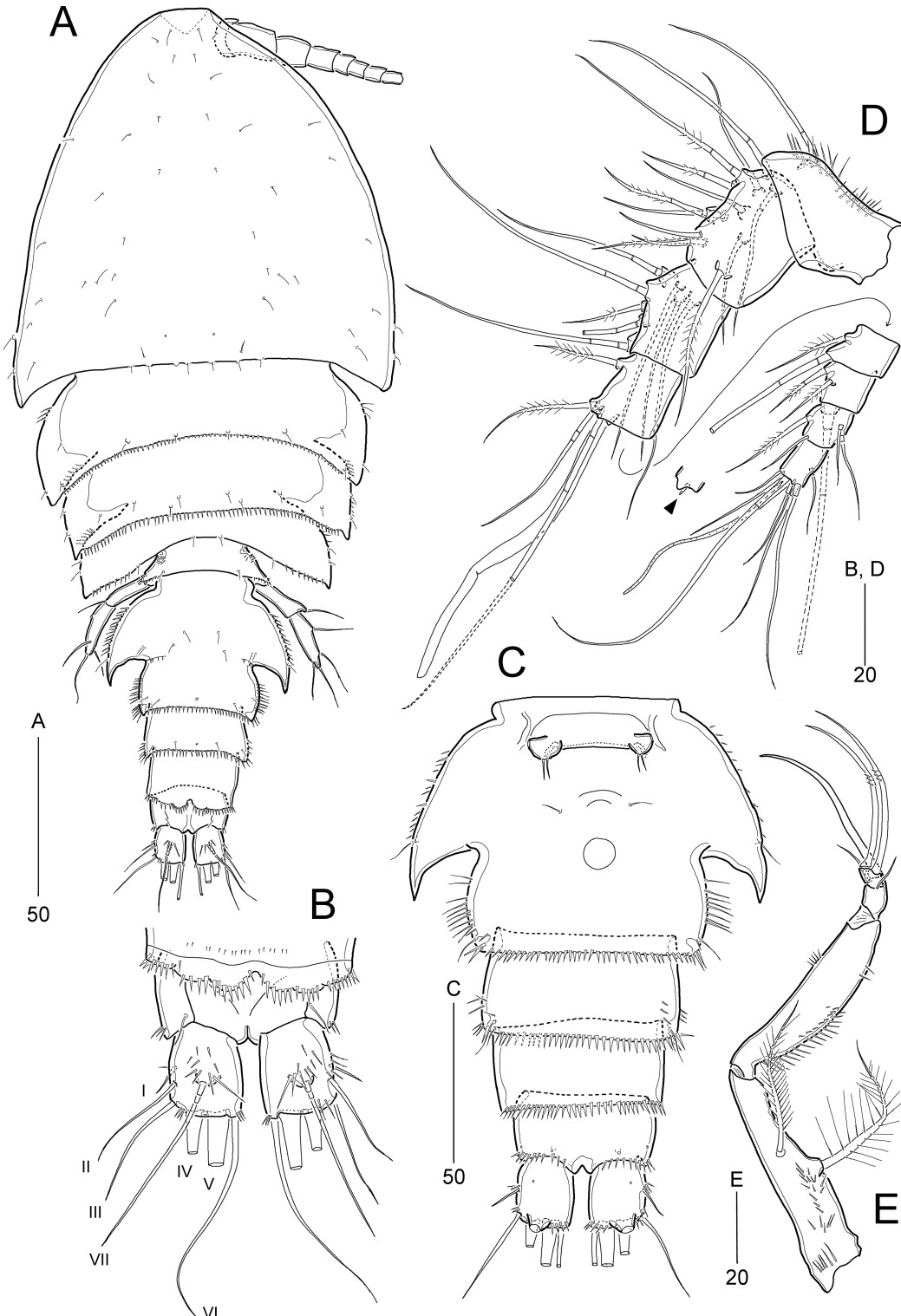

**Figure 6 *Idyella dolichi* sp. nov., female holotype (MABIK CR00257806).** (A) Habitus, dorsal; (B) Anal somite and caudal rami, dorsal; (C) urosome excluding P5 bearing-somite, ventral; (D) antennule, arrowhead indicating a rudimentary element on the distal segment; (E) maxilliped. Scale bars are in μm.

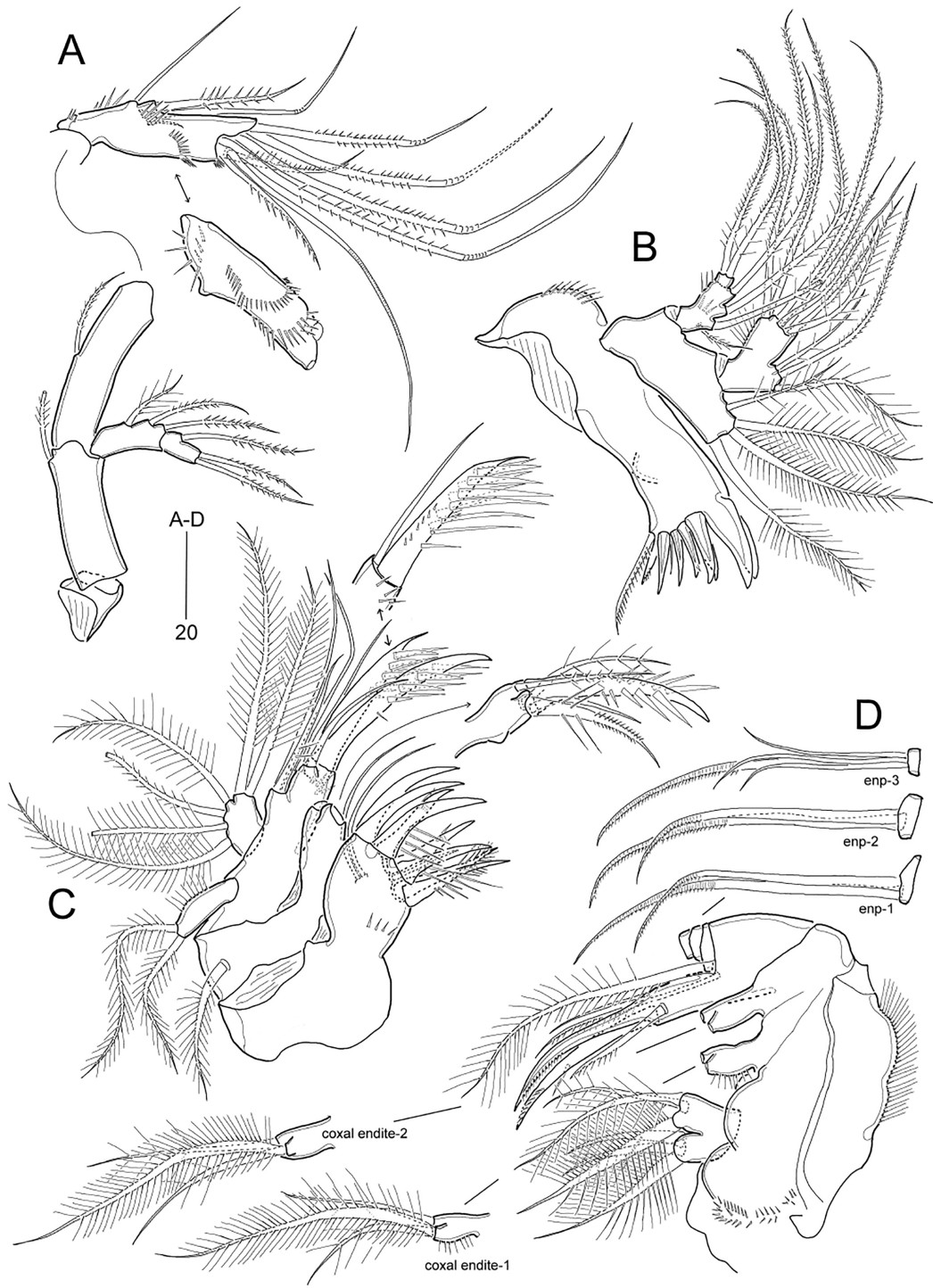

**Figure 7 *Idyella dolichi* sp. nov., female holotype (MABIK CR00257806).** (A) Antenna, second endopodal segment figured separately, with only surface ornamentation; (B) mandible; (C) maxillule, basal and coxal endites figured separately; (D) maxilla, two coxal endites and three endopodal segments figured separately. Scale bars are in μm.

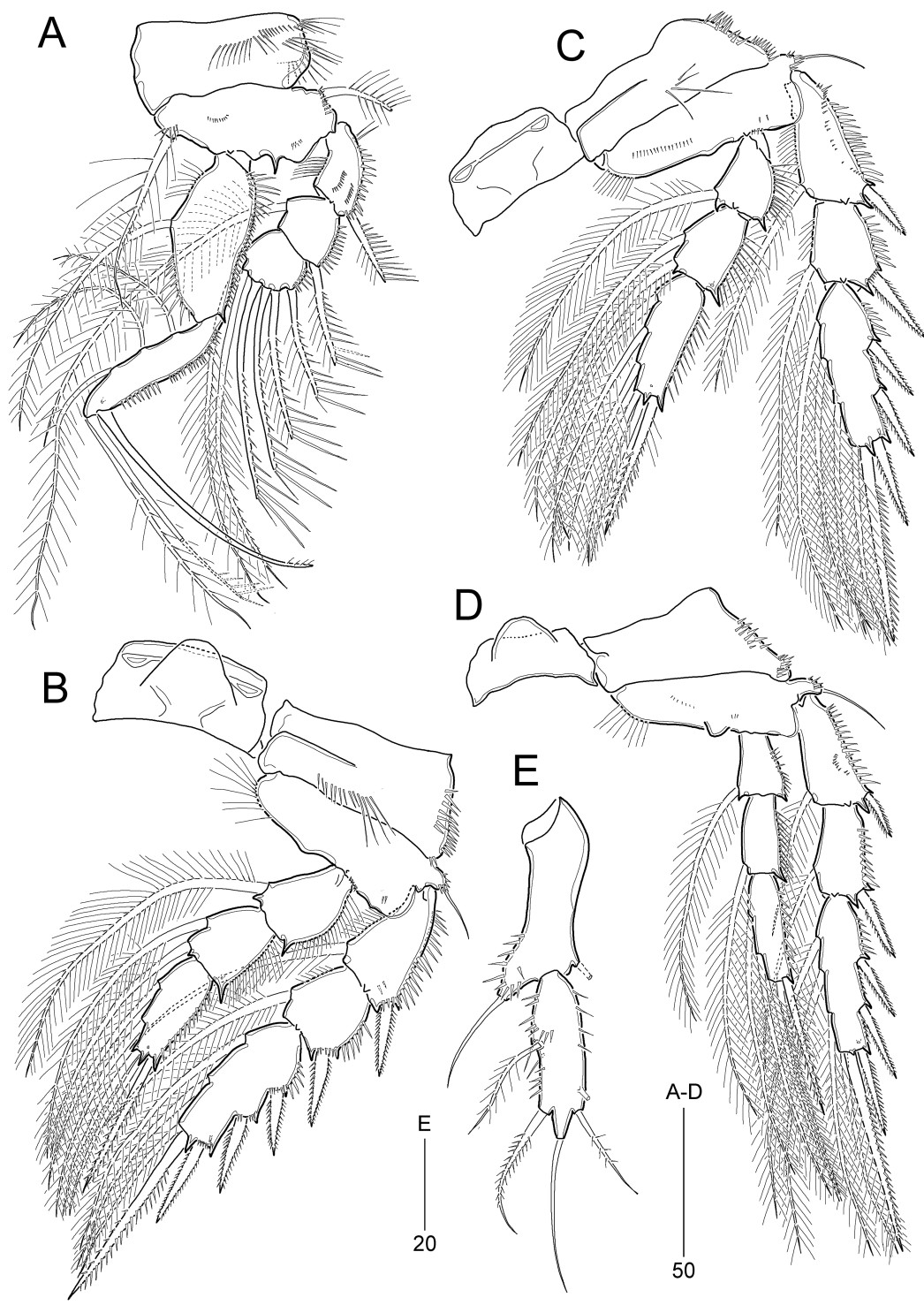

**Figure 8 *Idyella dolichi* sp. nov., female holotype (MABIK CR00257806).** (A) P1, anterior; (B) P2, anterior; (C) P3, anterior; (D) P4, anterior; (E) P5. Scale bars are in μm.

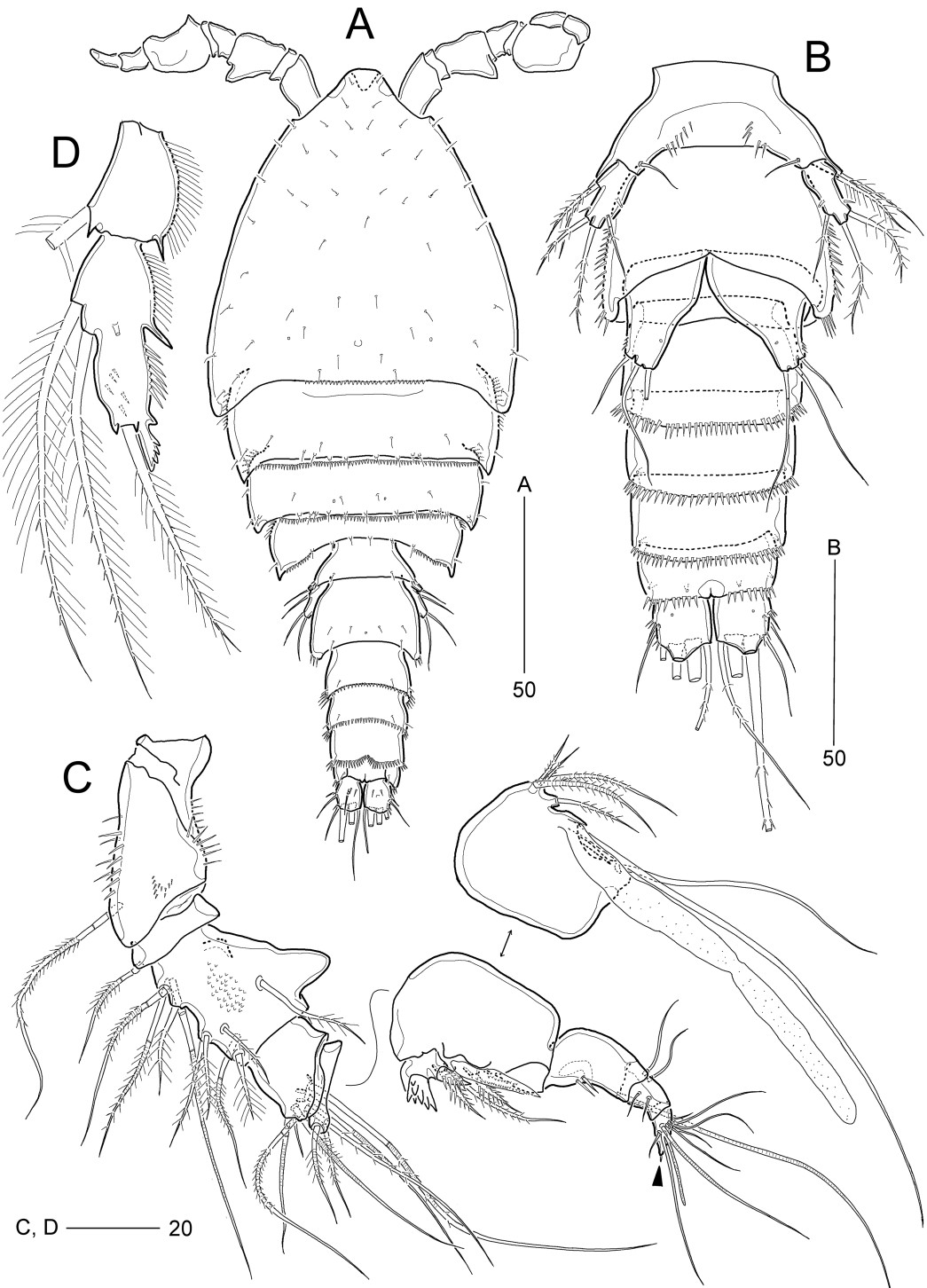

**Figure 9 *Idyella dolichi* sp. nov., male paratype (MABIK CR00257804).** (A) Habitus, dorsal; (B) urosome, ventral; (C) antennule, arrowhead indicating a rudimentary element, other side of sixth segment figured additionally; (D) P2 endopod, anterior. Scale bars are in μm.

genital slit covered on both sides by a small plate derived from sixth leg bearing one outer seta (lost during dissecting process) and two small distal setae; mid-ventral copulatory pore located at anterior 3/5 of genital double-somite accompanied by two sensilla. Penultimate somite with pseudoperculum with distal notch medially, ornamented with spinules. Anal somite (Fig. 6B) small, with deep dorsal incision, ornamented with spinules along posterior margin laterally and ventrally, one pair of dorsal sensilla, and two pairs of ventral pores; operculum invisible.

Caudal rami (Figs. 6B and 6C) slightly longer than wide, ornamented with group of dorsal spinules, few lateral spinules, two rows of distal spinules; ventral surface with one small pore and one subdistal peduncle bearing large pore; with seven setae: setae I and II inserted midway outer margin of ramus; seta I smallest, seta II longer than ramus (about three times as long as seta I); seta III slightly longer than seta II; setae IV and V (both setae damaged) not fused basally; seta VI inserted at inner distal corner, about three times as long as ramus; seta VII tri-articulate, inserted in proximal 2/3 of ramus on dorsal surface, slightly shorter than seta VI.

Rostrum (Fig. 6A) completely fused to cephalothorax, triangular, bent ventrally (invisible in dorsal view), with paired sensilla.

Antennule (Fig. 6D) 8-segmented, bent at outer border between first and second segments. First segment with one small outer protrusion and several rows of inner spinules. Fourth segment produced into pedestal, bearing one aesthetasc and one long seta (fused basally), at distal inner corner. Seventh segment small, with two bi-articulate outer setae, one bare and one plumose inner seta. Distal segment with three bare setae (of which distal one, indicated by arrowhead in Fig. 6D, rudimentary, spine-like), four bi-articulate setae, and one aesthetasc and one long seta (fused basally). Setal formula as follows: 1-[1], 2-[13], 3-[8], 4-[3 + (1 + ae)], 5-[2], 6-[3], 7-[4], 8-[7 + (1 + ae)].

Antenna (Fig. 7A) 4-segmented. Coxa small, unornamented. Basis elongate, with one abexopodal seta subdistally. Exopod 2-segmented; proximal segment elongate, about twice as long as distal exopodal segment, laterally with one plumose (proximal) and two (subdistal) pinnate setae; distal segment with one lateral and two distal pinnate setae. First endopodal segment longer than basis, with one pinnate seta midway inner margin; distal segment ornamented with five rows of spinules and two subdistal hyaline frills; lateral armature composed of one spinulose seta, one geniculate seta, and two bare (one small, one long) setae; distal armature comprising one bare seta, two uni-spinulose setae, of which outer fused to adjacent bare seta basally, and four spinulose and geniculate setae.

Mandible (Fig. 7B). Coxa with one small dorsal protrusion medially and one row of outer spinules proximally; gnathobase armed with two bi-cuspidate and three uni-cuspidate teeth, one uni-pinnate spine, one tiny spine and one uni-pinnate seta. Palp well-developed, biramous; basis broad, with four plumose distal setae, of which outermost shortest. Exopod imperfectly 2-segmented, one side completely fused, but other side with surface suture, and ornamented with two spinular rows; proximal segment elongated, with one small and three pinnate setae laterally; distal segment small, with two pinnate setae apically. Endopod broad, 1-segmented, with one pinnate and two plumose setae on lateral endite, and seven pinnate setae on distal margin.

Maxillule (Fig. 7C). Praecoxal arthrite well-developed, with one row of posterior spinules, two juxtaposed anterior setae, one bare distal seta, and eight stout distal spines (of which two anterior with one spinule, one anterior spinulose, and three posterior, with row of spinules proximally). Coxa with epipodite represented by one plumose seta; cylindrical endite with one pinnate seta and one stout and spinulose spine on lateral peduncle, and one plumose and one bare seta, and one spinulose spine on distal peduncle. Basis elongate, with two lateral endites; proximal endite with three bare setae, and distal endite with two stout and spinulose spines distally and subdistally and two bare lateral setae, with one row of anterior spinules. Exopod cylindrical, with two pinnate distal setae, and ornamented with inner spinules. Endopod small, broad, with six plumose setae.

Maxilla (Fig. 7D). Syncoxa large, ornamented with two paired spinular rows proximally and with one row of outer setules; with three endites: proximal endite bilobed, each lobe with three plumose setae; middle endite (first coxal endite) and distal endite (second coxal endite) cylindrical, each with one densely plumose and two weakly plumose setae distally. Allobasis distally with one plumose seta, one bare seta, and one uni-pinnate seta; inner part drawn out into a curved uni-pinnate claw accompanied by one uni-spinulose spine and one uni-pinnate seta. Endopod small, 3-segmented; first and second segments with two uni-pinnate (rat-tail-like subdistally) distal setae; distal segment with one uni-pinnate (rat-tail-like subdistally) and three bare distal setae.

Maxilliped (Fig. 6E) 4-segmented, comprising syncoxa, basis, and 2-segmented endopod. Syncoxa elongate, contracted in distal half, ornamented with several rows of proximal spinules; with two plumose setae. Basis as long as syncoxa, with two rows of spinules along slightly convex outer margin and with one row of spinules on almost linear inner margin. Endopod small, 2-segmented; proximal segment with one long and curved claw and two small and bare setae; distal segment smaller than preceding one, with two geniculate and uni-spinulose setae distally.

P1 (Fig. 8A). Intercoxal sclerite (not figured) well-developed, wide. Praecoxa (not figured) transversely elongated, triangular. Coxa large, with four rows of anterior spinules and one row of posterior spinules. Basis with two groups of anterior small spinules and one acute process on distal margin between exopod and endopod; outer seta plumose, arising from rudimental peduncle accompanied by small spinules basally; inner seta plumose basally and spinulose subdistally, accompanied by several spinules at its base, reaching distal end of enp-1. Exopod 3-segmented inwardly bent, reaching distal 1/3 of enp-1; each segment ornamented with outer spinules and exp-1 with two rows of anterior spinules and one row of inner setules; exp-1 and exp-2 each with one uni-spinulose outer spine; exp-2 with one long plumose inner seta; exp-3 with three outer spines, one outer distal spine and one inner distal seta, and one inner seta. Endopod 2-segmented; enp-1 broad inner margin convex, with one long plumose inner seta, one row of setules proximally and two rows of spinules subdistally, and outer margin ornamented with three rows of spinules; enp-2 as long as enp-1, with three plumose inner setae, two apical setae (one plumose, one uni-spinulose), one outer spine, and one anterior pore, and ornamented with two rows of outer spinules.

P2–P3 (Figs. 8B and 8C). Intercoxal sclerite well-developed, wide, unornamented. Praecoxa (not figured) transversely elongated, with spinules on anterior surface and on outer margin. Coxa with spinular rows on anterior surface and outer margin. Basis with one bare outer seta arising from peduncle with spinules at its base, ornamented with one or two rows of minute spinules on anterior surface, one row of setules along convex inner margin, and one row of minute spinules near base of endopod. Exopod 3-segmented, exopod longer than endopod; exp-1 ornamented with anterior spinules; exp-1 and exp-2 each ornamented with rows of outer spinules and one row of inner setules, and with one inner seta; exp-3 longest, with three pinnate outer spines, one pinnate outer apical spine and one plumose inner apical seta, and two (P2) or three (P3) plumose inner setae; P2 exp-3 with anterior tube pore near outer distal corner. Endopod 3-segmented; outer margins of enp-1 and enp-2 with setules, of enp-3 with spinules; enp-1 and enp-2 with one and two plumose inner setae, respectively; enp-3 longest, with one pinnate outer spine, two plumose apical setae, and two (P2) or three (P3) plumose inner setae; anterior pore present on enp-2 and enp-3.

P4 (Fig. 8D) smaller than other legs. Coxa with two groups of outer spinules on anterior surface. Basis ornamented as in P2 and P3, with small round process distally, and outer pedestal elongate, with one outer seta. Exopod 3-segmented; exp-1 and exp-2 as in P2 and P3 except for bare inner margins; exp-3 as in P3. Endopod 3-segmented, slenderer than those of P2 and P3; armature as in P2 except for enp-2 with only one plumose inner seta; outer margins ornamented with outer spinules, and enp-3 with one anterior pore subdistally.

Armature formula of P1–P4 given in Table 3.

P5 (Fig. 8E) 2-segmented. Baseoendopod elongated; outer pedestal produced, with one bare outer seta, ornamented with anterior spinules along margins; inner margin slightly swollen; endopodal lobe rudimental, with one seta (lost during dissecting process). Exopod slightly smaller than baseoendopod, about three times as long as broad, with two outer setae (one plumose, one pinnate), one bare apical seta (longest), and one pinnate inner seta, and ornamented with two rows of outer spinules and one row of inner spinules.

Description of male.—Body length 207.0 μm, measured from anterior margin of rostrum to posterior margin of caudal rami; habitus (Fig. 9A) largely as in female. Sexual dimorphism observed in urosomal segmentation, antennule, P2, P5, and P6.

Urosome (Figs. 9A and 9B) 6-segmented, ornamentation as figured; genital somite and first abdominal somite separate. Urosomites with 1–3 pairs of dorsal sensilla except for penultimate somite. Abdominal somites ornamented with small dorsal and large ventral spinules along distal margin except for anal somite. Anal somite and caudal rami as in female.

Antennule (Fig. 9C) 9-segmented, subchirocer, with geniculation between sixth and seventh segments. First segment ornamented with several spinular rows. Third segment with one strong outer process. Fifth segment smallest. Sixth segment swollen, with three modified elements (two proximal small spines serrated, one middle spine lanceolate, ornamented with denticles), segment with surface denticles. Distal segment conical,

**Table 3 Armature formula of P1–P4 of *Idyella dolichi* sp. nov.**

|  | Exopod | Endopod |
|---|---|---|
| P1 | 0.1.123 | 1.321 |
| P2 | 1.1.223 | 1.2.221 (1.21apo in ♂) |
| P3 | 1.1.323 | 1.2.321 |
| P4 | 1.1.323 | 1.1.221 |

slightly curved inwards, apically with very reduced element (indicated by arrowhead in Fig. 9C). Armature formula as follows: 1-[1], 2-[1], 3-[10], 4-[7], 5-[2], 6-[10 + three modified elements + (1 + ae)], 7-[3], 8-[4], 9-[8 + (1 + ae)].

P2 (Fig. 9D). Exopod as in female. Endopod 2-segmented; proximal segment as in female; distal segment formed by fusion of enp-2 and enp-3, with two plumose inner setae on original enp-2, and one plumose apical seta and one stout outer apophysis (fused to segment basally) on original enp-3 (Table 3); original enp-2 with one anterior tube pore.

P5 (Fig. 9B) baseoendopods fused medially and fused to somite; each leg with one outer basal seta and one spinular row; endopodal lobe represented by one bare seta. Exopod 1-segmented, about two times as long as wide, with one inner tube pore and four plumose setae (one inner, one apical and two outer); outer margin with one spinule.

P6 (Fig. 9B). Sixth pair of legs symmetrical, represented on both sides by subtrapezoidal plate with proximal inner margin convex; with one inner pore and three distal setae, and ornamented with outer spinules subdistally.

Etymology.—The name of the new species is derived from the ancient Greek word δολῐχή (*dolikhḗ*), alluding to the terminal endopodal segment of P1 possessing fairly long setae on the inner margin. Gender: feminine.

Remarks.—Within the family Idyanthidae, *Idyella* is the most species-rich genus with seven valid species: *Idyella australis* (Brady, 1910), *I. exigua* Sars, 1905, *I. kunzi* Bodin, 1968, *I. major* Sars, 1920, *I. nilmaensis* Kornev & Chertoprud, 2008, *I. pallidula* Sars, 1905 (type species), and *I. tenuis* (Brady, 1910). Earlier descriptions of them should be treated with caution when discerning species due to inadequate descriptions and illustrations. *Brady (1910)* illustrated the swimming legs of *I. australis* and *I. tenuis* but did not specify which leg was depicted. The setal armatures depicted in the Brady's figures suggest that they represent P3 in *I. australis* (see *Brady, 1910*: Textfig. LI., Fig. 6) and P4 in *I. tenuis* (see *Brady, 1910*: Textfig. L., Fig. 5). This is because P3 generally has a characteristic combination of three inner setae on the distal exopodal segment and two inner setae on the middle endopodal segment, whereas P4 has three inner setae on the distal exopodal segment and only one inner seta on the middle endopodal segment. If our view is true, *I. australis* is probably characterized by the presence of two inner setae on the terminal endopodal segment of the third leg, while the armature of *I. tenuis* is unique in that it lacks an inner seta on the proximal exopodal segment on the fourth leg. However, whether the inner seta of the P4 endopod was overlooked in the original description of *I. tenuis* requires further analysis. Similar cases are also observed in the middle segment of the P1 exopod

lacking inner seta in both species, as well as the absence of an outer spine on the two proximal segments of the P1 exopod in *I. australis*. *Brady*'s *(1910)* last observation is certainly incorrect, since it is a plesiomorphic armature complement of harpacticoids. On the other hand, in the original description of *I. tenuis*, *Brady (1910)* claimed that the species has a 9-segmented female antennule, but this statement could be due to an observational error since the 8-segmented condition hints at the position of an aesthetasc that is commonly present on the fourth segment in all other congeners.

*Kornev & Chertoprud (2008)* reported *I. nilmaensis* from muddy sediments at a depth of 40 m in the White Sea, with detailed illustrations of its morphology. Considering the common armature pattern of the thoracic legs, the figures of P3 (their Fig. 5.25И) and P4 (their Fig. 5.25З) were incorrectly mislabeled as P4 and P3, respectively.

Since there has been little close examination of the cephalosomal appendages for species delimitation in the genus *Idyella*, species identification has been primarily based on the female genital double-somite, P1 endopod, female P5, and caudal rami. According to the setation on the P5 exopod of the female, *Idyella* species can be readily subdivided into two groups. The first group has four setae on the segment and includes three species, *I. exigua*, *I. major*, and *I. pallidula*, whereas the second group has five setae and comprises three species, *I. kunzi*, *I. nilmaensis*, and *I. tenuis*; no information on this leg was given in the original description of *I. australis*. This subdivision is also supported by the presence or absence of a bending at the outer border between the first and second segments in the female antennules and it seems to be a synapomorphy for *I. exigua*, *I. major*, *I. pallidula*, and *I. dolichi* sp. nov.

The new species, *I. dolichi* sp. nov., is readily distinguished from the other three species of the first group, *i.e.*, the terminal segment of the P1 endopod has three setae (*vs.* only two setae in the other three species), the P5 baseoendopod has one seta on a rudimentary lobe (*vs.* two setae on a well-extended lobe in *I. exigua*), the genital double-somite has wing-like processes bearing a pointed posterior tip on the original genital somite similarly to that of *I. pallidula* (*vs.* wing-like processes present on the original genital, and the first and second abdominal somites in *I. exigua*; in *I. major*, wing-like processes are present only on the original genital, as in the new species, but the posterior tip is rounded and broad). In addition, fairly long inner setae on the terminal segment of the P1 endopod, which are as long as the terminal elements, can be used as a minor character for the specific justification of *I. dolichi* sp. nov.

Morphological information on sexual dimorphism in the genus *Idyella* is limited for *I. exigua*, *I. pallidula sensu Arlt, 1983*, and two undescribed species (*Idyella* spec. 1 and *Idyella* spec. 2) recorded by *Seifried (2003)*. The male P2 endopod of *I. dolichi* sp. nov. differs markedly from the typical form (see Remarks on the preceding species, *Idyellopsis orientalis* sp. nov.) in that the second and third segments are completely fused, and the middle hyaline seta is lacking, as in *I. exigua*. Furthermore, the uni-serrate condition of the modified spine in *I. dolichi* sp. nov. is unique among the known males of *Idyella* species. Additionally, the male of *I. dolichi* sp. nov. has a prominent outer projection on the third segment of the antennules. These morphological features in the male also support the establishment of *I. dolichi* sp. nov.

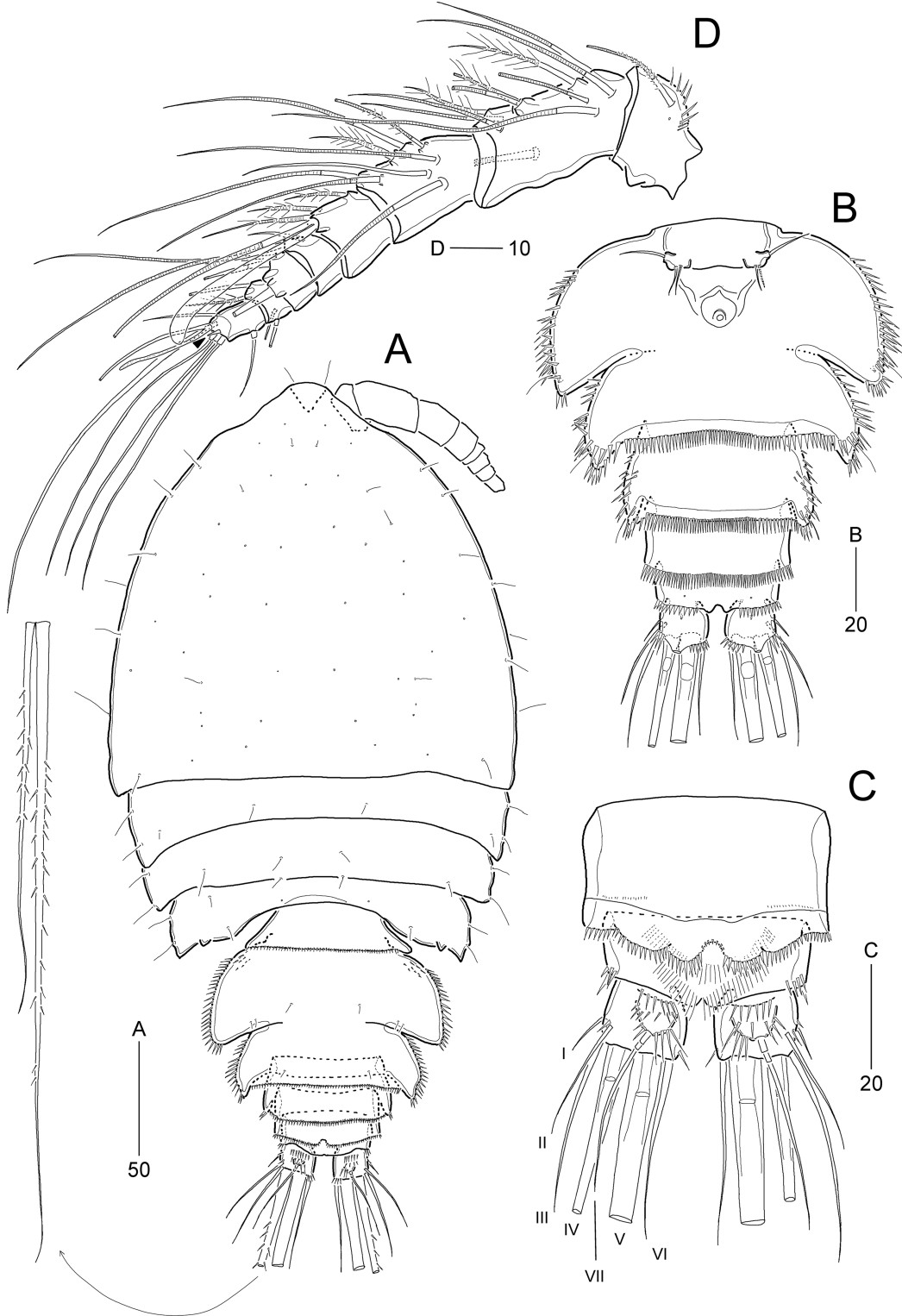

**Figure 10** *Idyella exochos* **sp. nov., female holotype (HNIBRIV12224, A) and paratype (HNIBRIV12225, B–D).** (A) Habitus, dorsal; (B) urosome excluding P5 bearing-somite, ventral; (C) penultimate somite, anal somite, and caudal rami, dorsal; (D) antennule, arrowhead indicating a rudimentary element. Scale bars are in μm. 

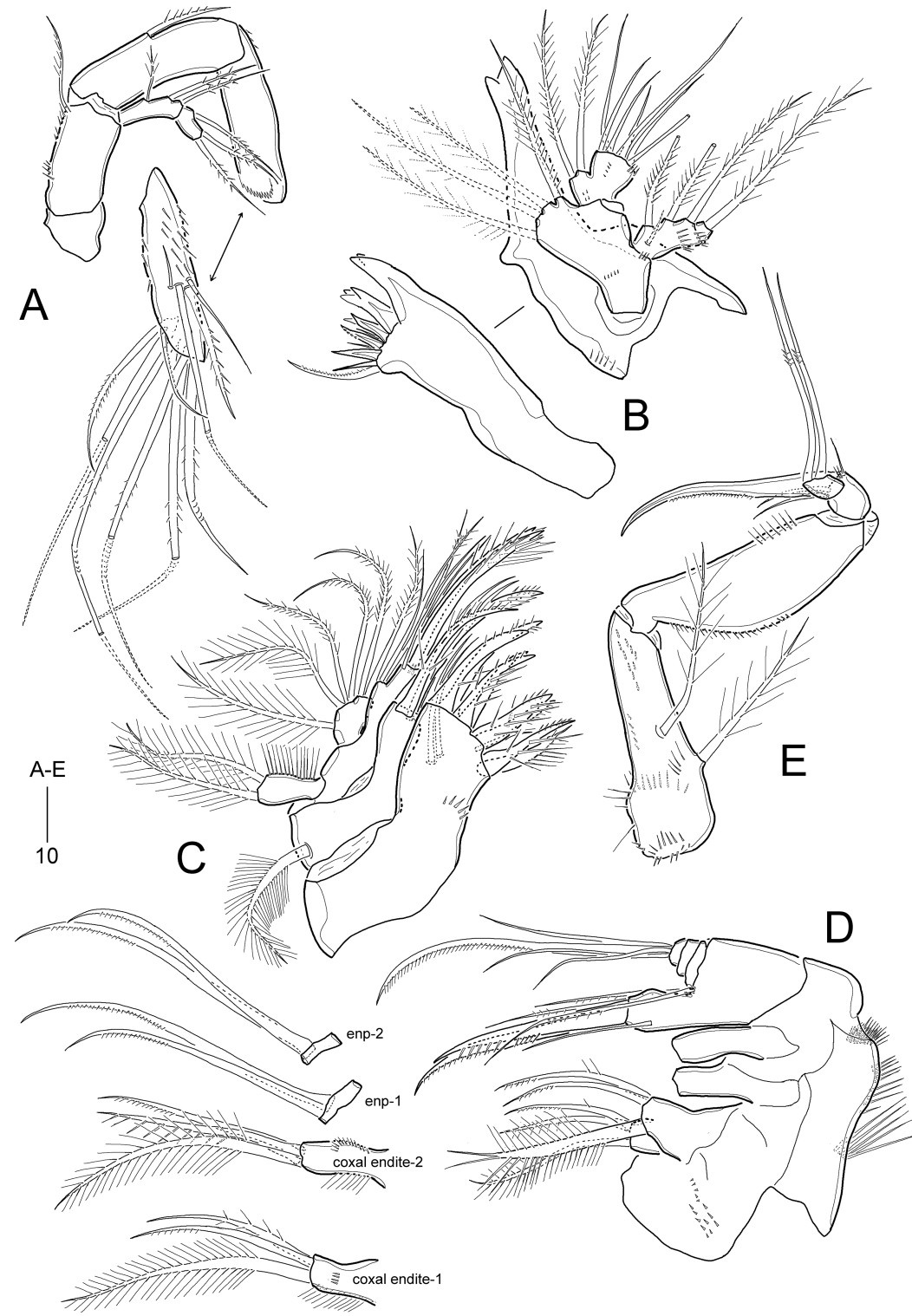

A-E

enp-2

enp-1

coxal endite-2

coxal endite-1

**Figure 11** *Idyella exochos* **sp. nov, female paratype (HNIBRIV12225).** (A) Antenna; (B) mandible; (C) maxillule; (D) maxilla, two coxal endites and two proximal endopodal segments figured additionally; (E) maxilliped. Scale bars are in μm.   

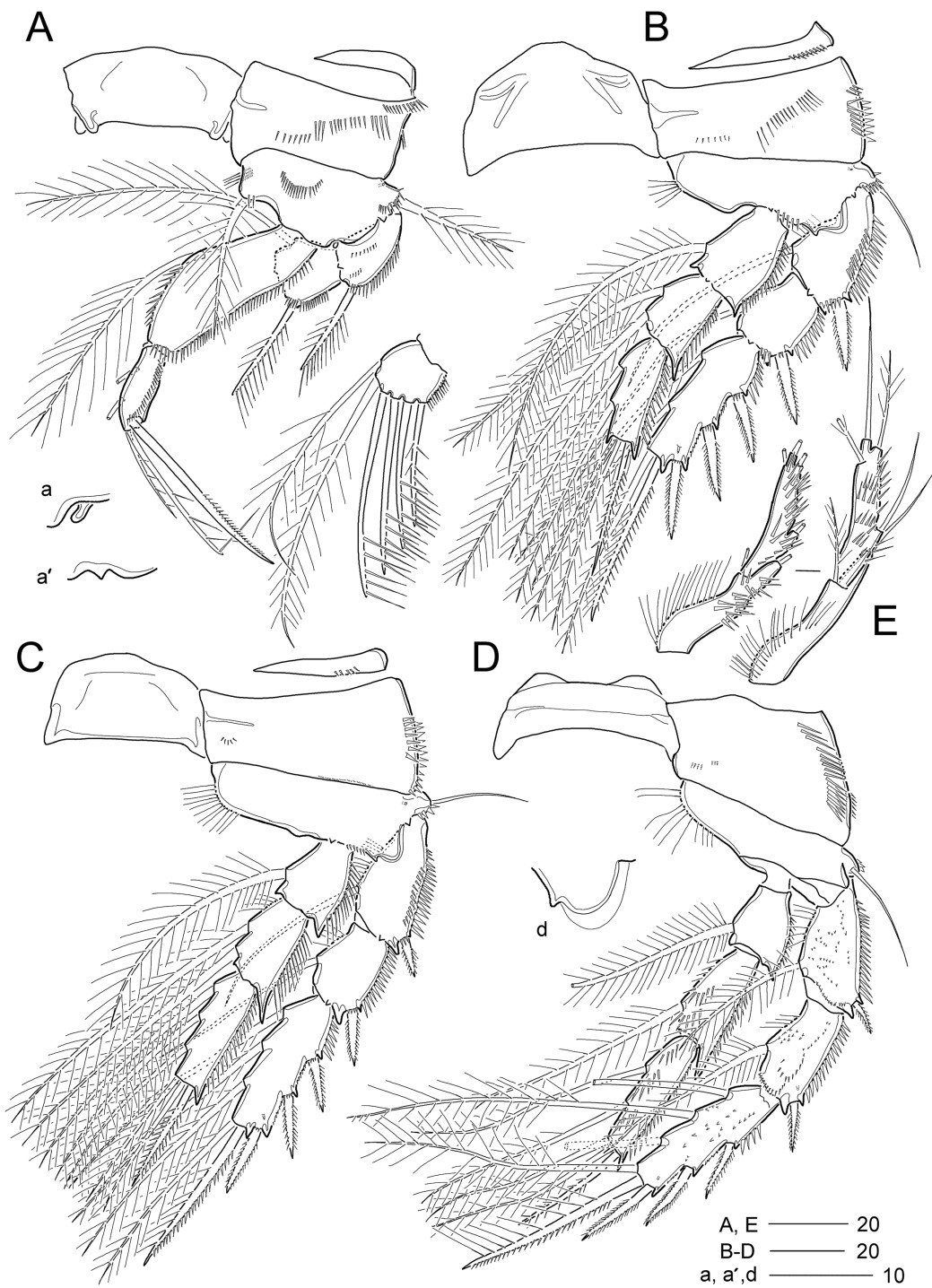

**Figure 12** *Idyella exochos* **sp. nov., female paratype (HNIBRIV12225, A, a, B–E) and male paratype (HNIBRIV12226, a′).** (A) P1, anterior, (a) a process on distal margin of basis in female, (a′) a process on distal margin of basis in male; (B) P2, anterior; (C) P3, anterior; (D) P4, posterior, (d) a process and lobe on distal margin of basis, anterior; (E) P5, other side figured additionally. Scale bars are in μm.

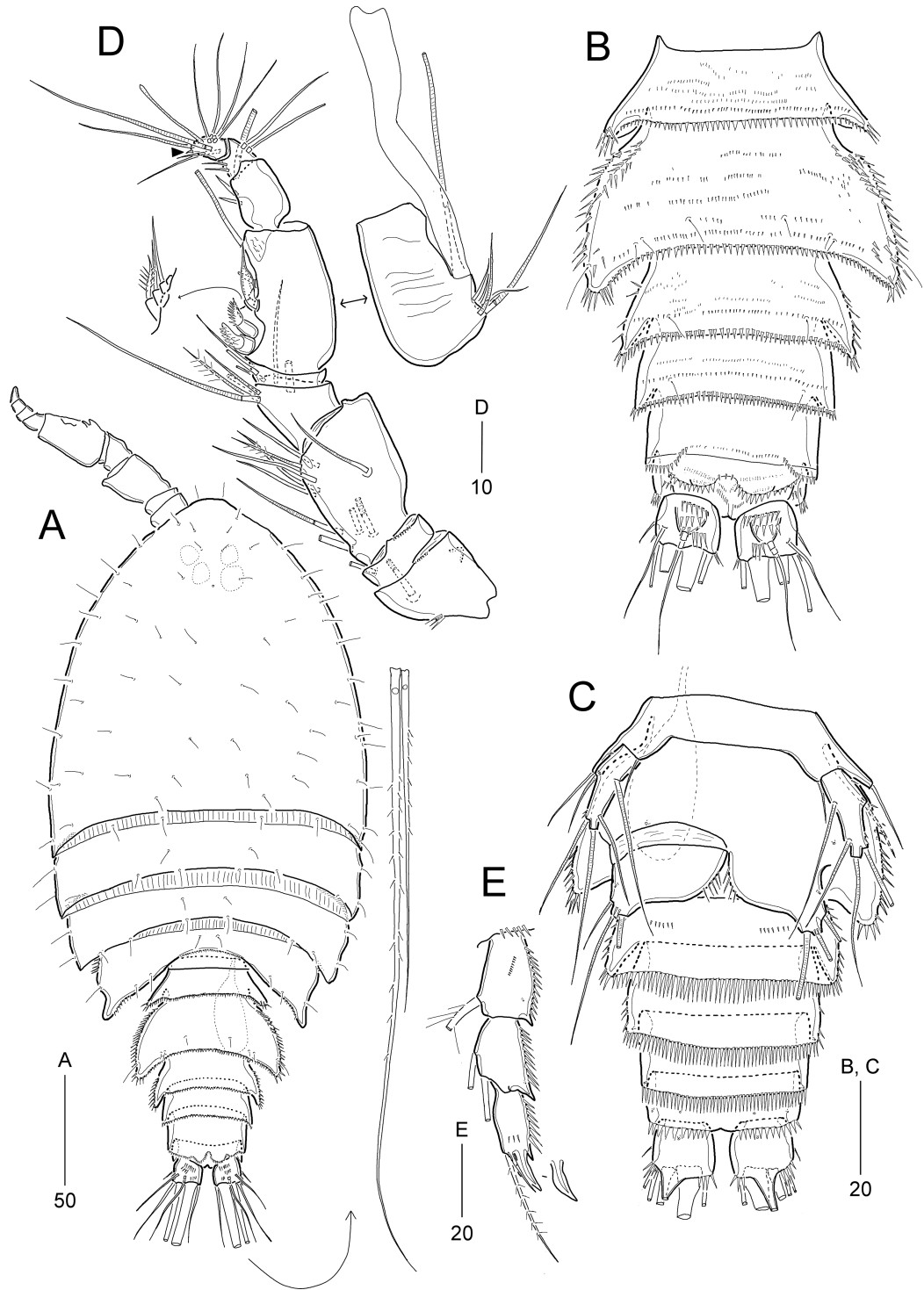

**Figure 13** *Idyella exochos* **sp. nov, male paratype (HNIBRIV12226).** (A) Habitus, dorsal; (B) urosome, dorsal; (C) urosome, ventral; (D) antennule, arrowhead indicating a rudimentary element, other side of sixth segment figured additionally; (E) P2 endopod, anterior, apical apophysis figured separately. Scale bars are in μm.

*Idyella exochos* sp. nov.
urn:lsid:zoobank.org:act: 2264448A-3E28-41AE-99E4-0D0284E00010
Figures 10–13

Type locality.—Dok-do Island, Korea, 37°14′56.39″N, 131°51′18.27″E, 55 m depth.

Type material.—Holotype: ♀ (HNIBRIV12224) preserved in a vial with 80% ethanol; collected from the type locality, 02 November 2018. Paratypes: 1 ♀ (HNIBRIV12225) dissected and mounted on eight H-S slides; 1 ♂ (HNIBRIV12226) dissected and mounted on seven H-S slides; 1 ♂ (HNIBRIV12227) preserved in a vial with 80% ethanol; collection data as in holotype; coll. J. G. Kim.

Description of female (based on holotype and paratype).—Holotype body length 347.9 μm (figured paratype: 422 μm) measured from anterior margin of cephalothorax to posterior margin of caudal rami in dorsal view.

Habitus (Fig. 10A) pyriform depressed, broader than that of *I. dolichi* sp. nov., with distinct separation between prosome and urosome. Prosome (Fig. 10A) longer than urosome, comprising cephalothorax and three free pedigerous somites. Cephalothorax broad, slightly wider than long, about 50% of body length. P4-bearing somite with one pair of papillae on posterior margin.

Urosome (Figs. 10A and 10B) 5-segmented, as in *I. dolichi* sp. nov., ornamentation as figured. P5-bearing somite smallest, with minute spinules along posterior margin in dorsal view. Genital double-somite completely fused, but with partial dorsolateral suture indicating original division; with spinular ornamentation along outer and posterior margins; pleural area well-developed, as wing-like processes on genital and first abdominal somites, anterior larger. Genital pores (Fig. 10B) separate, each covered by small plate bearing one outer and two posterior setae (representing P6); mid-ventral copulatory pore located at anterior 2/5 of genital double-somite. Penultimate somite with spinules along posterior margin and serrated at middle of pseudoperculum (Fig. 10C). Anal somite about three times as wide as long, with spinules along posterior margin laterally and ventrally, and two pairs of ventral pores; semicircular operculum ornamented with long fine spinules.

Caudal rami (Figs. 10B and 10C) slightly shorter than wide, with three groups of dorsal spinules; with seven setae: insertion of setae as in *I. dolichi* sp. nov.; setae I and II inserted in proximal 2/3 of ramus; seta I smallest, seta II 2.7 times as long as seta I, seta III about four times longer than seta I; setae IV and V not fused basally; setae IV and V bi-pinnate; seta V about two times longer than seta IV; seta VI as long as seta III; seta VII tri-articulate, inserted in proximal 2/3 of ramus on dorsal surface.

Rostrum (Fig. 10A) similar to *I. dolichi* sp. nov. (invisible in dorsal view).

Antennule (Fig. 10D) 8-segmented. First segment with one small outer process, one very small pore, and one row of inner spinules. Second segment longest, about twice as long as preceding segment. Fourth segment produced into distal pedestal bearing one aesthetasc and one long bare seta (fused basally). Seventh segment smallest, with two bi-articulate outer setae, and one bare and one plumose inner seta. Distal segment with three bare (of

which distal one very reduced) and four bi-articulate setae, and one aesthetasc and one long seta (fused basally). Setal formula: 1-[1], 2-[11], 3-[8], 4-[3 + (1 + ae)], 5-[2], 6-[3], 7-[4], 8-[7 + (1 + ae)].

Antenna (Fig. 11A). Coxa small, unornamented. Basis elongate, ornamented with two rows of inner spinules, and with one uni-plumose abexopodal seta subdistally. Exopod 2-segmented; proximal segment about twice as long as distal segment, with three pinnate lateral setae; distal segment with one lateral and two distal pinnate setae. First endopodal segment longer than basis, with one uni-pinnate medial inner seta; distal segment as long as preceding one, ornamented with several rows of spinules and one outer hyaline frill subdistally; lateral armature comprising one stout pinnate seta and three bare setae, of which outermost smallest; diagonal distal margin with three pinnate and four geniculate pinnate setae as in *I. dolichi* sp. nov., of which two adjacent outer apical setae fused basally.

Mandible (Fig. 11B). Coxa with one small process medially and one spinular row proximally; gnathobase armed with one tri-, two bi-, and three uni-cuspidate teeth, two spinules, and one uni-pinnate seta. Palp biramous; basis with four subequal distal setae. Exopod 2-segmented; proximal segment elongated, with one bare surface seta and three pinnate lateral setae, with two rows of spinules; distal segment very small, with one pinnate apical seta. Endopod 1-segmented, broad, with two plumose lateral setae and seven bare distal setae; ornamented with three groups of spinules.

Maxillule (Fig. 11C). Praecoxal arthrite with one row of posterior spinules, and two juxtaposed anterior setae; distal margin with one slender seta and eight spinulose spines. Coxa with one densely plumose epipodal seta; cylindrical endite distally with two spinulose spines and two bare setae. Basis elongate, with two endites; lateral endite with one bare and two plumose setae; distal endite with two bare setae and two spinulose spines and ornamented with few posterior spinules. Exopod elongate, ornamented with outer setules, and with two plumose distal setae. Endopod broad, with two plumose and four pinnate setae.

Maxilla (Fig. 11D). Syncoxa large, with one group of surface denticles proximally, and four rows of spinules along outer margin; with three endites: praecoxal endite bilobed, each lobe with one plumose and two pinnate setae distally; proximal coxal endite with one plumose and two weakly pinnate setae, ornamented with one row of lateral setules and one row of surface spinules; distal coxal endite cylindrical, with one plumose, one uni-spinulose and one pinnate seta, and ornamented with two rows of lateral setules. Allobasis with two pinnate setae proximally; endite with one bare seta laterally, one pinnate seta and two spinulose spines distally. Endopod small, 3-segmented; first and second segments with two uni-pinnate distal setae each; distal segment with one uni-pinnate and three bare setae distally.

Maxilliped (Fig. 11E) 4-segmented, comprising syncoxa, basis and 2-segmented endopod. Syncoxa elongate, tapering towards distal end, ornamented with several rows of various-sized spinules; with two plumose setae similar in length. Basis as long as syncoxa, unarmed; outer margin convex, ornamented with minute spinules in proximal half; inner margin almost linear, with one row of spinules in distal half. Endopod small, 2-segmented; proximal segment larger, with one stout claw and three bare setae, and ornamented with

few distal spinules; distal segment small, with two long geniculate setae bearing few spinules at geniculate regions.

P1 (Fig. 12A). Intercoxal sclerite well-developed, wide. Praecoxa transversely elongate, triangular, ornamented with few distal spinules. Coxa large, with one row of outer spinules and three rows of anterior spinules. Basis with one anterior pore, one row of anterior spinules, and one row of inner spinules; distal margin with one small rounded process between rami (Fig. 12a); outer pedestal with one plumose seta and ornamented with one row of anterior spinules; plumose inner seta reaching distal 3/4 of enp-1, flanked by few spinules at its base. Exopod as in *I. dolichi* sp. nov. Endopod longer than exopod, 2-segmented; enp-1 two times longer than enp-2, inner margin swollen in middle convex, ornamented with spinules in distal half and setules in proximal half, with one long plumose seta, and distal and outer margins ornamented with spinules; enp-2 cylindrical, with two inner setae, two uni-spinulose apical setae and one uni-pinnate outer spine, and ornamented with one row of outer spinules.

P2–P3 (Figs. 12B and 12C). Intercoxal sclerite well-developed, unornamented. Praecoxa transversely elongated, with one row of distal spinules. Coxa with spinular rows on anterior surface and outer margin. Basis with outer pedestal bearing one bare seta, with anterior spinules; anterior surface with one pore, inner margin convex, with one row of setules, and distal margin with one small acute process and one row of spinules between rami. Exopod 3-segmented, exopod slightly longer than endopod, exp-1 and exp-2 ornamented with spinular rows along outer margin as figured, and with one inner seta; P3 exp-1 and exp-2 ornamented with one row of inner setules; exp-3 longest, with three pinnate outer spines, one outer apical spine with pinnate outer and plumose inner margin, one plumose inner apical seta, and two (P2) or three (P3) plumose inner setae, and one anterior pore. Endopod 3-segmented, each segment with spinules along outer margin; enp-2 with two inner setae; P2 enp-2 with one anterior pore and P3 enp-2 with one row of inner setules; enp-3 longest, with one pinnate outer spine, two plumose apical setae, and two (P2) or three (P3) plumose inner setae.

P4 (Fig. 12D). Intercoxal sclerite well-developed, unornamented. Praecoxa (not figured) transversely elongated as in P2–P3. Coxa with one row of posterior spinules, one row of outer spinules, and two rows of anterior spinules. Basis with setules on inner margin; distal margin with one small acute process between rami (Fig. 12d); outer peduncle with long bare seta and ornamented with anterior spinules. Exopod 3-segmented, outer ornamentation as in *I. dolichi* sp. nov., covered with numerous posterior denticles; exp-1 and exp-2 each with one plumose inner seta; armature of exp-3 as in P3 exp-3. Endopod 3-segmented, ornamentation of outer and inner margins as in *I. dolichi* sp. nov.; enp-1 without posterior spinules; enp-2 and enp-3 each covered with posterior spinules; enp-1 and enp-2 each with one plumose inner seta; enp-3 with one pinnate outer spine, two plumose apical and two plumose inner setae.

Armature formula of P1–P4 given in Table 4.

P5 (Fig. 12E) 2-segmented. Baseoendopod elongate, with cylindrical outer pedestal bearing one bare seta; inner margin swollen in proximal half bearing setular

**Table 4 Armature formula of P1–P4 of *Idyella exochos* sp. nov.**

|  | Exopod | Endopod |
| --- | --- | --- |
| P1 | 0.1.123 | 1.221 |
| P2 | 1.1.223 | 1.2.221 (1.2.02apo in ♂) |
| P3 | 1.1.323 | 1.2.321 |
| P4 | 1.1.323 | 1.1.221 |

ornamentation and outer surface ornamented with several spinular rows; endopodal lobe completely incorporated into baseoendopod, represented by one plumose seta. Exopod about 3.7 times as long as broad, covered with several rows of outer spinules, with one plumose and two bare outer setae, one bare apical seta (longest), and one plumose inner seta.

Description of male.—Body length 307 μm; habitus (Fig 13A) largely as in female. Sexual dimorphism observed in urosomal segmentation, antennule, P1, P2, P5, and P6.

Urosome (Figs. 13A–13C) 6-segmented, ornamentation as figured. Urosomites with numerous rows of dorsal spinules. Abdominal somites ornamented with spinules along posterior margin ventrally. Anal somite and caudal rami as in female.

Antennule (Fig. 13D) 9-segmented, subchirocer, with geniculation between sixth and seventh segments. First segment with one small outer protrusion, one row of spinules proximally and one row of minute spinules distally. Second segment small, with one row of minute spinules distally. Third segment broad and longest. Fifth segment smallest. Sixth segment swollen, with three modified elements as in *I. dolichi* sp. nov. Distal segment slightly curved inwards, with one very reduced apical seta (indicated by arrowhead in Fig. 13D). Armature formula as follows: 1-[1], 2-[1], 3-[10], 4-[7], 5-[2], 6-[10 + three modified elements + (1 + ae)], 7-[3], 8-[4], 9-[8 + (1 + ae)].

P1 as in female, but distal margin of basis with one small, pointed process between rami (Fig. 12a′).

P2 (Fig. 13E). Exopod as in female. Endopod 3-segmented; enp-1 and enp-2 as in female, but enp-1 anteriorly with one pore and one row of spinules; enp-3 with one long plumose inner seta, one short apical seta, and one stout apical apophysis (Table 4); two apical elements fused basally to supporting segment.

P5 (Fig. 13C) with baseoendopods fused medially, and fused to somite as in *I. dolichi* sp. nov. Baseoendopod with one outer basal seta on cylindrical setophore; endopodal lobe completely incorporated into baseoendopod, represented by one plumose seta. Exopod 1-segmented, about 2.5 times as long as broad, with two bare outer, one bare apical and two bare inner setae; outer margin ornamented with few spinules subdistally.

P6 (Fig. 13C). Sixth pair of legs asymmetrical, represented on both sides by broad plate (fused to ventral wall of supporting somite on one side); ornamented with spinules along inner margin. Outer distal corner produced into short process bearing three bare setae, of which middle longest, and with few distal spinules.

Etymology.—The specific name *exochos* is derived from the Ancient Greek ἔξοχος (*éxokhos*), alluding to the presence of papillae (in the female) or projections (in the male) on the posterior border of the fourth pedigerous somites.

Remarks.—The new species, *I. exochos* sp. nov., also shares the armature of the tetrasetose female P5 exopod with *I. exigua*, *I. major*, *I. pallidula*, and *I. dolichi* sp. nov. (see Remarks of the preceding new species). Regarding the armature of the P1 endopod and female P5 baseoendopod, *I. exochos* sp. nov. appears to be similar to *I. major* and *I. pallidula* in the terminal segment of the P1 endopod and P5 baseoendopod with two inner setae and one seta, respectively. However, *I. exochos* sp. nov. distinctly differs from *I. major* and *I. pallidula* in that the length:width ratio (1.9) of the body is smaller in the former than in *I. major* (2.3) and *I. pallidula* (2.2), the terminal segment of the P1 endopod is markedly shorter than those of both species, the female genital double-somite has two pairs of wing-like processes, of which the posterior one is weakly developed or rudimentary in both species, and the P5 baseoendopod has a convex inner margin proximally compared to the straight inner margin in both species.

*Idyella dolichi* sp. nov. and *I. exochos* sp. nov. described here exhibit clearly different sexual dimorphic characteristics. In *I. exochos* sp. nov., the P2 endopod has the typical sexual dimorphic feature (see Remarks on *Idyellopsis orientalis* sp. nov.) of idyanthid copepods given by *Seifried (2003)*. The third segment of this ramus has three distal elements composed of a stout spine fused basally with the segment, a small hyaline seta, and a long plumose seta, while in *I. dolichi* sp. nov., the distal two segments are completely fused and the hyaline seta is lacking. The P5 exopod of the male of *I. exochos* sp. nov. is armed with five setae as in *I. exigua*, compared with four setae in *I. dolichi* sp. nov. In addition, the male of *I. exochos* sp. nov. has asymmetrical sixth legs, but they are symmetrical in *I. dolichi* sp. nov.

## DISCUSSION

Although the cladistic analysis conducted by *George (2023)* proposed a possible relationship among four genera of the subfamily Aspinothoracinae, the phylogenetic evaluation of other idyanthid genera within the family remains unclear. A morphological comparison of the genera *Idyella*, *Idyanthe*, *Tachidiella*, *Idyellopsis*, *Dactylopia* Becker, 1974, and *Nematovorax* suggests that they can be grouped into two distinct lineages based on the number of inner setae on the middle segment of the P3 endopod: the *Idyanthe–Idyella* lineage, which possesses only one inner seta, and the *Idyellopsis–Tachidiella–Nematovorax–Dactylopia* lineage, which bears two inner setae.

The sister-group relationship between *Idyanthe* and *Idyella* is supported by two synapomorphies: the presence of only one seta on the proximal segment of the antennary exopod and elongation of the female P5 exopod, as well as the presence of wing-like processes on the genital double-somite. Within this lineage, members of *Idyella* have two important autapomorphies: the P1 endopod is outwardly flexible at the junction between the proximal and distal segments, contrasting the prehensile and three-segmented endopod of *Idyanthe*; and the antennary exopod is two-segmented compared to the
three-segmented condition in *Idyanthe*. The prehensile condition can be considered an autapomorphy of *Idyanthe*. Insufficient descriptions of *Idyanthe* species hinder a detailed morphological comparison. The type species of *Idyanthe*, *Idyanthe dilatata* Sars, 1905, differs from its congeners in the armature of the female P5 baseoendopod, which has only one seta (*vs.* three setae in the other three species), and in the distal segment of the male P2 endopod, which features two modified apophyses fused to a supporting segment at the base (*vs.* one long apophysis and one seta in *Idyanthe australis* Pallares, 1970, and unknown in the other two species). *Idyanthe australis* retains the most primitive segmentation, with a nine-segmented female antennule and a two-segmented male P5 exopod (*Pallares, 1970*). Further comparisons of these character states will facilitate an assessment of the position and affinities of *Idyanthe* species.

To analyze the phylogenetic relationships among genera of the *Idyellopsis–Tachidiella–Nematovorax–Dactylopia* lineage, we compared the sexual dimorphism of the thoracic legs, as this is considered a robust indicator of phylogenetic relationships among harpacticoid taxa (*Huys, 1990*; *Huys & Kihara, 2010*). In Idyanthidae, male sexual dimorphism in the distal segment of the P2 endopod is characterized by four traits: the outer outgrowth is more developed than in females; the outer spine is confluent with the supporting segment; the outer apical element is greatly reduced in size; and there is a loss of two lateral setae that are homologous with the inner setae present in females (*Moura & Martínez Arbizu, 2003*; *Seifried, 2003*). However, males of *Idyellopsis orientalis* sp. nov. and *Idyellopsis typica* deviate from the typical form of the P2 endopod, possessing two inner setae on the distal segment (Fig. 5D). This observation appears to represent a plesiomorphic character state compared with other idyanthid genera, indicating the basal position of *Idyellopsis* within the family. *Bröhldick (2005)* hesitated to place *Nematovorax* in the family Idyanthidae due to its retention of many plesiomorphic character states of Idyanthidimorpha (*Seifried, 2003*). *Nematovorax gebkelinae* Bröhldick, 2005 shares two primitive character states, a nine-segmented female antennule and a three-segmented antennary exopod, with two *Idyellopsis* species, but has the typical P2 endopod lacking inner elements on the terminal segment. The *Tachidiella–Dactylopia* clade can be grouped by the outwardly curved distal segment of the P2 endopod, a modification likely formed by fusion of the outer apophysis (spine) to the segment. Species of *Tachidiella* retain many plesiomorphic states, including a distinct rostrum and three-segmented P1 endopod, whereas *Dactylopia peruana* Becker, 1974 has numerous autapomorphic characters, such as a rudimentary endopod of the maxillule, a one-segmented endopod of the maxilliped, reduced segmentation of the endopods in P3 and P4, the middle segment of the P1 exopod lacking an inner seta, and reduced armature of P2–P4, which include bare outer spines on all exopodal segments.

*Moura & Martínez Arbizu (2003)* suggested that the genera *Styracothorax* Huys, 1993 and *Aspinothorax* should be added to the family Idyanthidae, formally synonymizing the deep-sea family Styracothoracidae with the family Idyanthidae (*Seifried, 2003*). This phylogenetic relationship was primarily supported by the male sexual dimorphism of the distal segment of the P2 endopod (*Moura & Martínez Arbizu, 2003*: 180). This taxonomic revision led subsequent researchers (*e.g.*, *George, 2004*; *George & Wiest, 2015*) to place

tentatively two other deep-sea genera, *Meteorina*, and *Pseudometeorina*, in the family, although they could not provide phylogenetic evidence for their assignment. Notably, *Pseudometeorina mystica* George & Wiest, 2015, the type species of the genus, deviates significantly from five autapomorphic characteristics of the family (*cf. Seifried, 2003*: 93; *George & Wiest, 2015*: 577; *George, 2023*: 35). Thus, the taxonomic position of this deep-sea group of idyanthid genera remained somewhat ambiguous. In a recent phylogenetic study based on detailed morphological comparisons, *George (2023)* defined the subfamily Aspinothoracinae to encompass the genera *Aspinothorax*, *Pseudometeorina*, *Meteorina*, and *Styracothorax*, justified by five unambiguous autapomorphic characters: a two-segmented P1 exopod and transverse elongation of the bases of P1–P4 (*George, 2023*: characters 12–16).

Most harpacticoid copepods have adapted to a benthic lifestyle *via* morphological modifications designed for burrowing or crawling. The ability to crawl on sediment surfaces or other substrates among organisms such as macroalgae and mollusk shells is primarily achieved through a wide depressed body shape, thoracic legs characterized by short, wide, enlarged protopods, and slender, shortened rami, as well as the ability of the thoracic rami to bend inward (*Heptner & Ivanenko, 2002*). The transverse elongation of the bases of P1–P4 is a derived feature of members of the subfamily Aspinothoracinae. *Huys & Boxshall (1991)* described this feature as an adaption of thoracic legs to mud sediments, referring to it as a "spider-like habitus". This adaptation likely enhances crawling ability by providing space for the folding rami. Similar morphological adaptations can be observed in some deep-sea members of other harpacticoid families, such as Ancorabolidae Sars, 1909 and Cletodidae Scott, 1904, as well as in the phytal families Peltidiidae Claus, 1860 and Tegastidae Sars, 1904. Although these families have the characteristic rami suggested by *Heptner & Ivanenko (2002)*, the thoracic legs of the subfamily Aspinothoracinae represent a swimming–crawling type, which suggests that this morphological adaptation is associated with an epibenthic lifestyle on deep-sea muddy bottoms, and that this lineage has evolved independently within the subfamily.

## CONCLUSIONS

This study is the first record of the harpacticoid family Idyanthidae in Korean waters, based on the discovery of three new species: *Idyellopsis orientalis* sp. nov., *Idyella dolichi* sp. nov., and *Idyella exochos* sp. nov. Although a comparative analysis with the genera *Idyellopsis* and *Idyella* is restricted due to the scarcity of detailed morphological information in earlier descriptions, the evidence supporting our proposals for these three new species is sufficient in the following aspects: *Idyellopsis orientalis* sp. nov., which is a second representative of the so far monotypic genus *Idyellopsis*, differs from *Idyellopsis typica* in the length:width ratio of the body and in the genital double-somite, length of the caudal seta I, and structure of the female P5 baseoendopod; *Idyella dolichi* sp. nov. is characterized by the combination of the armature of the distal segment of the P1 endopod bearing three inner long setae, the female P5 bearing one and four setae on the baseoendopod and exopod, respectively, the presence of an outer protrusion on the third segment in the male antennule, and the fusion of the second and third segments in the

male P2 endopod; *Idyella exochos* sp. nov. shares the armature of the female P5 with *Idyella dolichi* sp. nov., but is distinguished from its congeners by the broad body, relatively short terminal segment of the P1 endopod bearing two inner setae, and the presence of two pairs of wing-like processes on the female genital double-somite.

The family Idyanthidae is one of the smallest groups in Harpacticoida, but many as yet undescribed species are waiting to be described (*e.g.*, *Seifried, 2003*; *Baguley, 2004*; *Shimanaga, Kitazato & Shirayama, 2004*; *Kitahashi et al., 2013*, *2014*; *George, Pointner & Packmor, 2018*). The discovery of three new species in the present study led us to assume a wide geographical distribution of the family, or at least *Idyellopsis* and *Idyella*, and their high species diversity in Korean waters. Further attention to their taxonomy might provide new insights into the species diversity and phylogeny of the family, as well as of the Idyantidimorpha (Idyanthidae plus Zosimeidae).

A morphological comparison reveals two lineages in idyanthid copepods, excluding the aspinothoracin harpacticoids: the *Idyanthe–Idyella* and *Idyellopsis–Tachidiella–Nematovorax–Dactylopia* lineages. The loss of an inner seta on the middle segment of the P3 endopod is a synapomorphy for the latter lineage. The genus *Idyella* has a more derived condition than *Idyanthe* in the P1 endopod, characterized by its outward flexibility. The genus *Idyellopsis* likely occupies a basal position within the latter lineage, considering the plesiomorphic character states of the P2 endopod in males.

## ACKNOWLEDGEMENTS

We would like to thank the captains and crews of the RV ONNURI, EARDO and JANGMOK of the Korea Institute of Ocean Science and Technology (KIOST) for sampling of sublittoral sediments. The authors also acknowledge Dr. S. L. Kim (KIOST) for assistance in sampling during a cruise on board RV EARDO.

### Funding

This work was financially supported by the management of Marine Fishery Bio-resources Center (2024) funded by the National Marine Biodiversity Institute of Korea (MABIK) and by the research program of the Korea Institute of Ocean Science & Technology (Contract No. PEA0202) for discovering *Idyellopsis orientalis* sp. nov and *Idyella dolichi* sp. nov. This study was also conducted with the support through research programmes (HNIBR 202101101 and HNIBR202301105) offered from the Honam National Institute of Biological Resources (HNIBR) for establishing *Idyella exochos* sp. nov. The funders had no role in study design, data collection and analysis, decision to publish, or preparation of the manuscript.

### Grant Disclosures

The following grant information was disclosed by the authors:
National Marine Biodiversity Institute of Korea (MABIK).
Korea Institute of Ocean Science &Technology: PEA0202.

Honam National Institute of Biological Resources (HNIBR): HNIBR 202101101 and HNIBR202301105.

## Competing Interests

The authors declare that they have no competing interests.

## Author Contributions

- Jong Guk Kim performed the experiments, analyzed the data, prepared figures and/or tables, authored or reviewed drafts of the article, and approved the final draft.
- Kyuhee Cho performed the experiments, prepared figures and/or tables, and approved the final draft.
- Jimin Lee conceived and designed the experiments, authored or reviewed drafts of the article, and approved the final draft.

## Data Availability

The descriptions and illustrations of three new species are available in the text and figures.

## New Species Registration

The following information was supplied regarding the registration of a newly described species:

Publication LSID: urn:lsid:zoobank.org:pub:311543E0-7EBE-4FE5-BC2A-C56B047172A2

*Idyellopsis orientalis* sp. nov.: urn:lsid:zoobank.org:act:9998A5DB-A356-43A8-8628-5AFBD9E369EE

*Idyella dolichi* sp. nov.: urn:lsid:zoobank.org:act:5487A79B-3A11-4622-82AD-0A731A91CEBF

*Idyella exochos* sp. nov.: urn:lsid:zoobank.org:act:2264448A-3E28-41AE-99E4-0D0284E00010

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
