# Peer review of "Three new species of the family Idyanthidae (Copepoda, Harpacticoida) from sublittoral zones around the Korean Peninsula"

_PeerJ, doi:10.7717/peerj.18767_

## Round 0.1 · original submission · Major Revisions

Dear Dr. Kim,

All reviewers find your article interesting. However they raise some concerns and I ask you to provide a point-by-point reply to each comment, particularly focusing on the location and characterization of the obtained samples. Also reviewer # 2 provided a very important feedback with an annotated version of the manuscript, thus please consider it.

Best regards

Reviewer 1 ·

Basic reporting

The manuscript is well-written. Professional English is used throughout, but some parts can be improved. I suggest adopting a more telegraphic style for the descriptions. Some suggestions are indicated in the revised PDF.
The authors showed ample experience with the studied taxa. They cited the most relevant literature. However, I suggest checking the authority and date of some taxa mentioned in the text. The manuscript is well-structured, but I suggest adding a new generic diagnosis for Idyellopsis. This is important given that, before the find of I. orientalis, this was a monotypic genus, whose diagnosis was based solely on the type and only species, I. typica. A differential diagnosis for the three new species would also be good. I suggest checking the gender of the species names...the species names might need to agree in gender with the name of the genera. The figures are of excellent quality. I suggest, however, checking the legends of some figures so everything is clear. The authors presented sound results in agreement with the hypotheses.

Experimental design

The content of the manuscript is original, and will be of great help for researchers working on several disciplines related with the systematics and taxonomy of harpacticoid copepods.
The research questions are clear, and the authors identified unambiguously the knowledge gaps that shall be filled with their research.
The investigation and its methods is rigurous and conforms to the provisions of the ICZN, but some comments were added to the revised PDF.
The methods were described with sufficient detail, and I only suggest replacing one reference (Huys et al. (1996) for Lang (1934)).

Validity of the findings

This manuscript contains valuable information for systamatists and taxonomists working on harpacticoid copepods. The discovery and description of three new species is also a valuable source of information to better understand the marine biodiversity of Korea.
The authors provided all the data and information to understand their point of view regarding the establishment of three new species of harpacticoid copepods.
The authors presented sound conclusions based on the results of their research.

Additional comments

No comment.

Annotated reviews are not available for download in order to protect the identity of reviewers who chose to remain anonymous.

Reviewer 2 ·

Basic reporting

no comment

Experimental design

no comment

Validity of the findings

no comment

Additional comments

The authors present a detailed and important description of 3 new species of the Idyanthidae (Copepoda, Harpacticoida) from sublittoral zones around the Korean Peninsula. In general, the manuscript is comprehensible, well-written as far as I am concerned (no English native speaker) and and worth of being published in PeerJ. However, I detected several potential slips of the pen (please see attached PDF).
Although Abstract and Introduction are informative and satisfying quality, I would like to make a few points: The authors mentioned in the Abstract that there are few studies on the family Idyanthidae in the Pacific Ocean. I would have liked to see a brief summary of these studies in the Introduction to help the reader understand the distribution of the taxon in the Pacific Ocean. As far as I know, the genus Idyellopsis has already been recorded from some localities in the Pacific Ocean such as Sagami Bay and Kuril Trench.
The Materials and Methods is also acceptable in general. However, not providing a map with sampling stations is a deficiency. When I plotted the coordinates on the map, I noticed that the stations were located both in the east and west of the Korean Peninsula. It is quite difficult for the reader to understand location of the stations by just reading the coordinates.
The Results is understandable and provides an adequate description of the new species. The illustrations are of very good quality and fulfill absolutely current standards.
The conclusions make me hesitant about the MS. The definition of a new species involves formulating a new hypothesis that deserves (requires!) an excellent justification. In the case of taxonomy/systematics, this justification is the phylogenetic discussion. Unfortunately, the authors do not provide such a phylogenetic discussion in this MS. Instead, they offer a very brief and superficial comparison of the new species with other Idyella, and they rely only on typological aspects without any phylogenetic relationships. I would have expected a more detailed comparison, which would have resulted in a phylogenetic hypothesis (shown, for instance, in a cladogram) of the relationships within Idyella. So the whole conclusions/discussion is a typological and simply phenetic comparison of characteristics. In my opinion, this is unacceptable. Therefore, I propose a revision of the whole conclusion/discussion, which means that in my opinion the paper is acceptable for publication in PeerJ only after a major revision.

Annotated reviews are not available for download in order to protect the identity of reviewers who chose to remain anonymous.

Reviewer 3 ·

Basic reporting

This is an excellent paper making an important contribution to the taxonomy of a lesser known family of deep-water harpacticoid copepods. The research is well executed, the text is well written including a historical overview of the genera the authors are dealing with. Most importantly, the illustrations are excellent which is always a pleasure to see as it forms the basis of taxonomic descriptions - a text description is always an accessory repetition of what is already illustrated. The authors have provided all the details in their excellent illustrations, have followed ICZN rules and cited all the relevant literature. The manuscript reports on a significant range extension of the Idyanthidae and provides a thorough discussion substantiating the identity and authenticity of the new species. I highly recommend this manuscript for publication in PeerJ.

Experimental design

No comment

Validity of the findings

The establishment of the new species is fully justified based on the detailed descriptions that were provided. The authors have provided sound evidence in their discussion that the taxa they discovered are new.

Additional comments

No comment

---

## Round 0.2 · Minor Revisions

Dear Dr. Lee,

Thanks for the resubmission of your manuscript.
Please carefully address the remaining comments of the single reviewer.

Best regards
Rodrigo

Reviewer 1 ·

Basic reporting

This is the second revision of this manuscript. Like the first version, the second revised version of this manuscript is well written. The authors used clear and unambiguous professional English throughout.
The authors showed ample knowledge on the taxonomy of the family Idyanthidae.
This manuscript is well-structured, the tables are very informative, and the line drawings are of excellent quality.
The proposal of the new species is relevant and will be useful for future systematic studies on this family.
The authors did a great job and incorporated new information. They added a generic diagnosis for Idyellopsis and a brief discussion on the phylogeny of the family. The generic diagnosis will be very useful in future studies, but I suggest some amendments so that the authors incorporate the characters and character states of I. typica and I. orientalis.

Experimental design

This manuscript deals with the description of three new species of benthic harpacticoid copepods of the family Idyanthidae from Korea, and the phylogenetic relationships amongst its genera. This contribution will be a valuable source of information for an in depth knowledge on the diversity of poorly known taxa of this region.
The author's investigation meets current standards in the field of taxonomy and systematics of harpacticoid copepods. The research question is clear and the results are relevant to unveil the true diversity of small metazoans from Korea.
The methods are described in detail, and the conclusions are supported by the results.
The authors followed the provision of the ICZN.
I detected some minor typographical errors and some corrections were done to the main document.

Validity of the findings

No comment.

Additional comments

No comment.

Annotated reviews are not available for download in order to protect the identity of reviewers who chose to remain anonymous.

Reviewer 2 ·

Basic reporting

No comment

Experimental design

No comment

Validity of the findings

No comment

Additional comments

The revised version of the manuscript has been improved considerably and is generally apt to be published in PeerJ. The authors did a good work!

---

## Round 0.3 · accepted · Accept

Congratulations on the acceptance of your manuscript.

Reviewer 1 ·

Basic reporting

No comment.

Experimental design

No comment.

Validity of the findings

No comment.

Additional comments

This is the third round of revisions. The authors did a good job and heeded the recommendations made on this occasion.